# ANCHOR FRAME BRIDGING FOR COHERENT FIRST-LAST FRAME VIDEO GENERATION

Xuehan Hou[1] [*]   Meng Fan[1,4*]   Pengchong Qiao[1]   Zesen Cheng[1]   Yian Zhao[1]
Lei Zhu[1]   Kaiwen Cheng[1,4]   Chang Liu[5†]   Jie Chen[2,3,1,4†]

[1] School of Electronic and Computer Engineering, Peking University, Shenzhen, China
[2] School of Intelligence Science and Engineering, Harbin Institute of Technology, Shenzhen, China
[3] Pengcheng Laboratory, Shenzhen, China
[4] AI for Science (AI4S)-Preferred Program, Peking University Shenzhen Graduate School, China
[5] Department of Automation and BNRist, Tsinghua University, Beijing, China

## ABSTRACT

First-last frame video generation has recently gained significant attention. It enables coherent motion generation between specified first and last frames. However, this approach suffers from semantic degradation in intermediate frames, causing scene distortion and subject deformation that undermine temporal consistency. To address this issue, we introduce Anchor Frame Bridging (AFB), a novel plug-and-play method that explicitly bridges semantic continuity from boundary frames to intermediate frames, offering training-free adaptability and generalizability. By adaptively interpolating anchor frames at temporally critical locations exhibiting maximal semantic discontinuities, our approach effectively mitigates semantic drift in intermediate frames. Specifically, we propose an adaptive anchor frame selection module, which generates text-aligned candidate frames via frame order reversal and selects anchors based on semantic continuity. Subsequently, we develop anchor frame-guided generation, which leverages the selected anchor frames to guide semantic propagation across intermediate frames, ensuring consistent boundary semantics and preserving temporal coherence throughout the video sequence. The final video is synthesized using the first frame, last frame, selected anchor frames, and the text prompt. The results demonstrate that our method significantly enhances the temporal consistency and overall quality of generated videos. Specifically, when applied to the Wan2.1-I2V model, it yields improvements of 16.58% in FVD and 10.21% in PSNR.

## 1 INTRODUCTION

Video generation models have shown remarkable proficiency in generating high-quality videos and support multiple generation paradigms, including text-to-video generation Hong et al. (2023); Villegas et al. (2023); Singer et al. (2023); OpenAI (2023); Zhang et al. (2024); Li et al. (2025); Yang et al. (2025b), image-to-video generation Ho et al. (2022); Blattmann et al. (2023); Peng et al. (2024); Yang et al. (2025a); Wang et al. (2025b); Namekata et al. (2025),etc. However, the first-last frame video generation Wang et al. (2025a); Bytedance (2024) remains a valuable yet underexplored area, which involves creating a video that simulates consistent motion between the first and last frames. Given its unique characteristics, an effective first-last frame video generation method can serve as a powerful tool in many applications, such as video editing, physical modeling, etc.

Although it is certainly possible to train large-scale video generation models to accept specific frames as conditions, training such a model requires too much time and economic cost. Most existing methods Bytedance (2024); Wang et al. (2025a) realize first-last frame video generation by reusing image-to-video models, which use both first and last frames as control conditions. Despite some success, these methods still suffer from information attenuation in intermediate frames. That is, the deterministic semantics of the first and last frames gradually decrease as they propagate to the

---

[*]Equal contribution.
[†]Corresponding author.

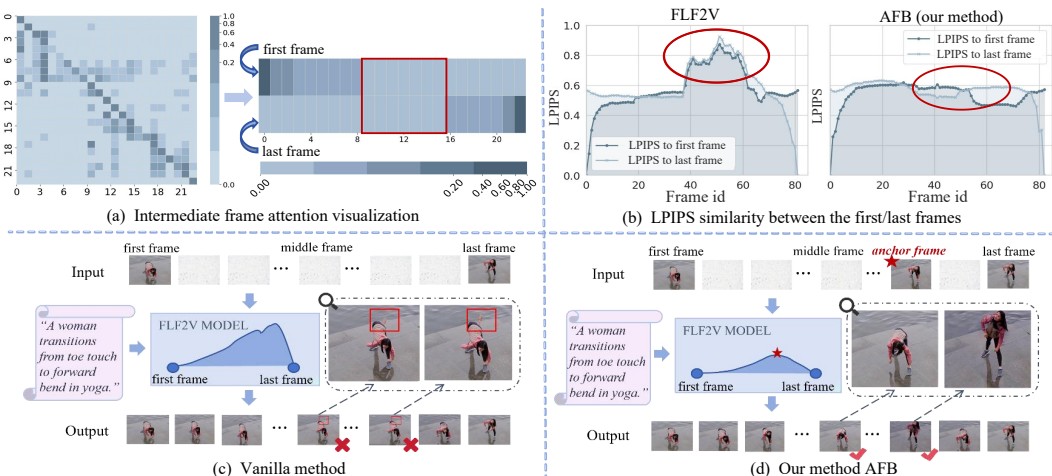

Figure 1: **Anchor Frame Bridging (AFB) design.** (a) Visualization of inter-frame attention in video generation via DiT's self-attention (left). Visualization of attention from the first-last frames to each intermediate frame (right). Notably, only adjacent frames exhibit substantial inter-frame attention values, while the first and last frames show relatively low attention to the middle frames, suggesting that the deterministic semantics in the first-last frames gradually diminish as information flows toward the intermediate frames. (b) Under the original method, LPIPS similarity between the first and last frames drops suddenly in middle frames. Our AFB shows smooth similarity change without such a drop. (c) Vanilla method's generated videos exhibit poor middle-frame quality. (d) Our AFB method smooths middle frames and enhances video temporal consistency.

intermediate frames. This is because, in self-attention layers, only adjacent frames have significant inter-frame attention values, causing lower attention weights from the first/last frames to the intermediate ones, as shown in Fig.1(a). This results in poor temporal consistency in the final video. As illustrated in Fig.1(b), frames near the first frame show higher generation consistency, while those in the mid-to-late part of the sequence show reduced coherence. Moreover, to maintain continuity with the terminal frame, the last few frames abruptly adopt the ending frame's attributes, causing abrupt transitions and unstable temporal dynamics throughout the video sequence.

To this end, we propose a novel **Anchor Frame Bridging (AFB)** method to bridge the semantics of the first and last frames to the intermediate frames. By strategically interpolating anchor frames at points of continuity breaks in generated videos, this method effectively reduces mid-sequence frame deviations and enforces controlled trajectory alignment throughout the generation process, as shown in Fig.1(d). Our AFB operates through two core steps: adaptively selecting an appropriate anchor frame from a candidate set, and then utilizing it to guide the generation of the final video.

Specifically, the **adaptive anchor frame selection** module is used to obtain anchor frames. As the low quality of intermediate frames adversely affects subsequent frames, the initial video segments maintain higher quality, whereas coherence breakdowns typically occur in the mid-to-late sections. To address this issue, we reverse the order of the first and last frames, thereby obtaining a higher-quality candidate set for anchor frame selection. We detect temporal discontinuities in the generated video and denote the position of the most significant break as $\alpha$. Empirically, such breakpoints occur at nearly symmetric relative positions in forward and reverse generation (Appendix E). Accordingly, we select the anchor at the mirrored position $1 - \alpha$.

After obtaining the anchor frame, we present the **anchor frame guided generation** to guide the generation process using the first frame, last frame and anchor frame. The anchor frame is inserted at the $\alpha$ position of the video. For the given frames (first, last and anchor frames), we apply a binary mask and concatenate features from the first/last frames with textual features before feeding them into the model. For other frames, we directly feed them into the model. This approach ensures the temporal consistency and coherence of generated video maintains.

Given the lack of datasets for first-last frame video generation, we first construct a dataset comprising 436 pairs of first and last frame images along with corresponding text prompts. We then conduct both qualitative and quantitative analyses on this dataset. We test our method on two common open source

I2V models: Wan2.1 and Hunyuan Video and conduct comparative experiments alongside Wan2.1-FLF2V Wang et al. (2025a), ViBiDSampler Yang et al. (2025a), and Generative Inbetweening Wang et al. (2025b). The results show that our method consistently outperforms the original baselines, achieving improvements of 16.58% in FVD and 10.21% in PSNR compared to the Wan2.1-I2V. Our main contributions are as follows:

- We introduce Anchor Frame Bridging (AFB), a method that bridges semantic continuity between first-last frames through anchor frame interpolation, mitigating intermediate distortion and resolving temporal inconsistency in first-last frame video generation.
- AFB is a plug-and-play method that operates in two stages: adaptive anchor frame selection via temporal continuity analysis and anchor frame-guided generation for coherent videos.
- We provide a high-quality first-last frame video generation dataset and conduct extensive experiments on it. Results show that our method significantly outperforms existing methods, achieving 16.58% improvement in FVD and 10.21% in PSNR.

## 2 RELATED WORK

**Image-to-Video Diffusion Models.** Image-to-Video Generation (I2V) focuses on generating dynamic video sequences from input images guided by text prompts. This approach enhances the controllability of video generation by anchoring the output to a specific first frame. Existing methods, such as I2VGen-XL Zhang et al. (2023), SVD Blattmann et al. (2023), Hunyuan Video-I2V Weijie Kong et al. (2024), CogVideoX Yang et al. (2025b) and Wan2.1-I2V Wang et al. (2025a) extend the text to video (T2V) framework to I2V by concatenating conditional latent representations with noise latents. These methods leverage the robust prior knowledge embedded in the underlying T2V models to achieve high-quality I2V generation. However, the incorporation of this prior knowledge tends to render the generated videos more reliant on text rather than input images. Frames initially resemble the input image, but gradually drift, which does not satisfy user requirements.

**Video Frame Interpolation.** Video frame interpolation aims to generate appropriate intermediate frames from two boundary frames. Flow-based methods Park et al. (2020); Kong et al. (2022); Huang et al. (2022); Li et al. (2023); Hu et al. (2024); Zhang et al. (2025) explicitly estimate optical flow or leverage flow guidance but struggle with complex extreme motions. Diffusion-based methods have also been explored in several studies. Voleti et al. (2022); Danier et al. (2023); Jain et al. (2024); Shen et al. (2024). Some methods employ a time-reversal strategy that fuses forward and reverse denoising paths to ensure adherence to both first and last frames. However, when the first and last frames differ significantly, the forward and backward paths exhibit substantial dynamic discrepancies, causing undesirable artifacts in the fused video frames. To mitigate such discrepancies, existing methods often introduce additional costs, such as multiple noise reinjections Yang et al. (2025a), frame-level constraints Zhu et al. (2025) and extra model training Wang et al. (2025b). This leads to significant computational overhead, compromising the system's reasoning efficiency. In contrast, our method AFB, which leverages an adaptive anchor frame addition strategy, transmits the information of the first and last frames to the intermediate frames. This eliminates the need for complex fusion steps and yields more coherent results.

**First-Last Frame to Video.** First-Last Frame Video Generation (FLF2V) is a novel video generation task, which involves automatically generating coherent intermediate frames from given first-last frames and text prompts, enabling highly controllable and consistent video generation. Make Pixels Dance Zeng et al. (2024) first proposed a diffusion-based model architecture, which is capable of generating videos guided by text and first-last frames, employing a strategy of concatenating the first and last frames with zero-padding to create dynamic videos. Earlier this year, Wan2.1-FLF2V Wang et al. (2025a) released a 14B-parameter FLF2V model, which concatenates the first and last frames with noise and feeds them into the diffusion model via a conditional control branch. Current FLF2V methods, fine-tuned from I2V models, suffer from weakened first-last frames semantics in intermediate frames. To mitigate this problem, we propose the Anchor Frame Bridging (AFB) to explicitly enhance the propagation of the semantics of the first and last frames, thereby significantly improving the consistency and video quality of first-last frame video generation.

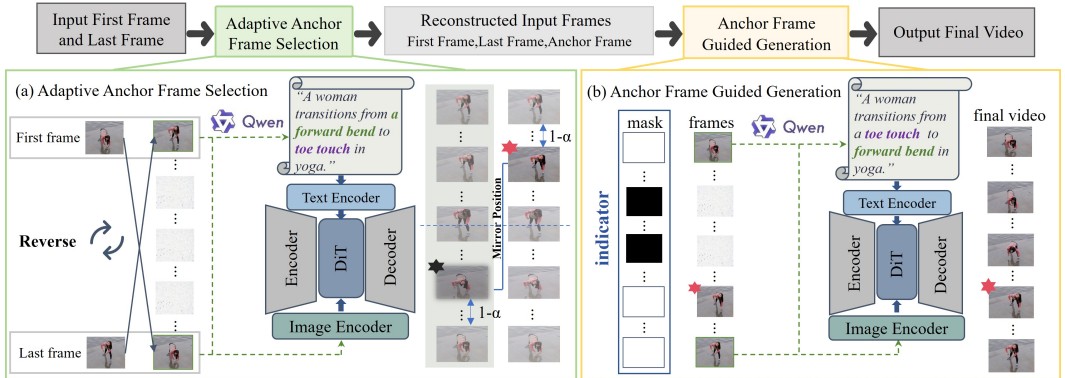

Figure 2: **Method architecture.** (a) Adaptive Anchor Frame Selection: Text prompts generated by Qwen, along with the swapped first and last frames, are fed into the video generation model to produce the anchor frame candidate set. Then we use the adaptive anchor frame selection method to select the anchor frame. (b) Anchor Frame Guided Generation: The first frame, last frame and anchor frame are concatenated with zero-padded frames along the time axis. The indicator guides the model's input conditions and specifies which frames will be generate. After denoising and decoding, we obtain the final video with high consistency to the first and last frames.

## 3    METHOD

To address the information attenuation in intermediate frames, we propose AFB to incorporate the semantic information from the first and last frames into the middle part of the video by inserting anchor frames. As illustrated in Fig. 2, our method consists of two main components. The adaptive anchor frame selection module first generates a candidate set of anchor frames and selects the most suitable one, which is then utilized by the anchor frame guided generation module to produce the final video. Detailed descriptions of these two modules are presented in the subsequent sections.

### 3.1    PRELIMINARIES

Diffusion models learn the underlying data distribution $x_0 \sim q(x_0)$ by learning to reverse a predefined, iterative process of noising data. For I2V diffusion models Blattmann et al. (2023), the process often operates in the latent space of a pre-trained VAE encoder to reduce computational complexity, where a clean video sample $x_0 \in \mathbb{R}^{N \times H \times W}$ is first encoded into latent space as $z_0 = E(x_0)$. A forward Markov process is then applied to gradually corrupts this latent representation into pure Gaussian noise over $T$ steps:

$$q(\boldsymbol{z}_{1:T} \mid \boldsymbol{z}_0) = \prod_{t=1}^{T} q(\boldsymbol{z}_t \mid \boldsymbol{z}_{t-1}), \quad q(\boldsymbol{z}_t \mid \boldsymbol{z}_{t-1}) = \mathcal{N}\left(\boldsymbol{z}_t; \sqrt{1-\beta_t}\boldsymbol{z}_{t-1}, \beta_t\mathbf{I}\right), \quad (1)$$

where $\beta_t \in (0, 1)$ is a predefined noise schedule at time step $t$. The reverse process reconstructs a clean latent from pure noise $\boldsymbol{z}_T$ via iterative denoising, which can be formulated as:

$$\boldsymbol{z}_{t-1} = \text{update}(\boldsymbol{z}_t, f_\theta(\boldsymbol{z}_t; t, \boldsymbol{c}); t), \quad (2)$$

where $f_\theta$ is a denoising network, and $c$ is the conditioning signal. In some flow-based model, the network is specifically parameterized as $u_\theta$ to predict a velocity field. After the iterative denoising, the final clean latent is decoded back to the generated video via a VAE decoder.

### 3.2    ADAPTIVE ANCHOR FRAME SELECTION

To obtain anchor frames with high similarity to the first and last frames, we design the adaptive anchor frame selection module, which consists of two core stages, as shown in Fig. 2(a). First, a reverse generation process is employed to create an anchor frame candidate set. Second, we apply an adaptive anchor frame selection strategy to select the appropriate anchor frame from this set.

**Reverse Generation Process.** Given the first frame $I_0$ and last frame $I_{N-1}$, we swap their positions and employ Qwen Bai et al. (2023) to generate a reverse-direction prompt denoted as $P^{\text{rev}}$, which captures the semantic and visual context of both frames and explicitly describes the transition dynamics between them,

$$P^{\text{rev}} = \text{Qwen}(I_{N-1}, I_0). \tag{3}$$

This text prompt is subsequently encoded into a text embedding, denoted as $c_{P^{\text{rev}}}$. The new initial frame $I_{N-1}$ and final frame $I_0$, are encoded to form the conditional input $z_c$:

$$z_c = E(I_{N-1}, I_0), \tag{4}$$

where $E$ is the VAE encoder.

The conditional input $z_c$, along with text embedding $c_{P^{\text{rev}}}$, guides the model during sampling. The iterative denoising update can be formulated as

$$z_{t-1} = \text{update}(z_t, u_\theta(z_t; t, z_c, c_{P^{\text{rev}}}); t), \tag{5}$$

where $u_\theta(z_t; t, z_c, c_{P^{\text{rev}}})$ denotes the output velocity predicted by the model, and $t$ represents the timesteps.

We terminate the reverse denoising at a stop step $K$. Since the latent $z_t$ is noise-corrupted at intermediate timesteps, we compute the predicted clean sample $\hat{z}_0$ derived from the current step $z_t$:

$$\hat{z}_0 = \frac{z_t - \sqrt{1 - \bar{\alpha}_t}\epsilon_\theta(z_t, t)}{\sqrt{\bar{\alpha}_t}} \tag{6}$$

Decoding the predicted $\hat{z}_0$ at stop step $K$ yields a sequence of predicted clean frames $\{I_n\}_{n=0}^{N-1}$, which forms the candidate set for anchor selection. Let $T$ be the total denoising steps, we perform reverse denoising for $K \leq T$ steps. $K = T$ corresponds to the full reverse process, whereas $K < T$ provides an accelerated variant.

**Adaptive Anchor Frame Selection.** To identify a suitable anchor frame from this candidate set and determine its insertion position, we first define a frame quality evaluation function $Q$ to assess the generation quality of each frame $I_n$ in the candidate set. We evaluate the entire sequence to find the frame with the poorest quality, located at index $n_p$:

$$n_p = \arg\min_n Q(I_n), \tag{7}$$

where $n \in [0, N-1]$. This index corresponds to the normalized relative position $\alpha = n_p/(N-1)$, which marks the location of the most significant *continuity breakpoint*.

As discussed in Appendix E, the *continuity breakpoint* location is nearly symmetric between forward and reverse generation. To mitigate this information attenuation at the corresponding position in the forward process, we replace the generated frame at location $\alpha$ with a high-quality anchor frame selected from the candidate set obtained via reverse generation. Since the reverse generation process swaps the first and last frames, the index of the anchor frame in the candidate set $n_a$ should follow a mirror positional relationship with $\alpha$:

$$n_a = (N-1)(1-\alpha). \tag{8}$$

The frame $n_a$ is usually close to the first frame (the original last frame) of the reverse generation, which means it is less affected by information attenuation, and typically exhibits high quality and strong semantic alignment. These characteristics make it an ideal anchor frame, denoted as $I_a$.

In this work, we measure frame quality based on LPIPS. Unlike metrics such as PSNR and SSIM, which perform pixel-level image comparison, LPIPS utilizes a pre-trained deep neural network to more accurately simulate human visual perception. A higher LPIPS value indicates a greater perceptual difference between adjacent frames, corresponding to poorer coherence in the video segment. Therefore, the peak values of LPIPS mark the transition points with the worst temporal coherence in the video sequence. Our final frame quality evaluation function $Q$ is defined as the negative of the local average LPIPS for frame $I_n$:

$$Q(I_n) = -\frac{1}{2}(\text{LPIPS}(I_{n-1}, I_n) + \text{LPIPS}(I_n, I_{n+1})). \tag{9}$$

Thus, a large LPIPS value results in a small $Q(I_n)$, signifying a low quality for the frame $I_n$.

## 3.3 Anchor Frame Guided Generation

After obtaining the anchor frame, we feed the first frame $I_0$, last frame $I_{N-1}$ and anchor frame $I_a$ as conditions into the video generation model as shown in Fig.2(b). To inform the model of the three frames we already have and prevent their regeneration, we design an indicator that specifies which frames need generation and which are already available. We leverage the reasoning capability of Qwen to derive a coherent text prompt $P^{\text{fwd}}$ from the given first and last frame images:

$$P^{\text{fwd}} = \text{Qwen}(I_0, I_{N-1}). \tag{10}$$

Subsequently, we extract the image features from the first-last frames and the text features $c_{P^{\text{fwd}}}$ from the generated prompt $P^{\text{fwd}}$. After combining these features, they are fed into the video generation model along with the indicator to guide the final video generation.

Different I2V models have different sampling methods. In this paper, we use the Wan2.1-I2V to illustrate the specific implementation of our anchor frame guided generation module. In Wan2.1-I2V, we use an additional binary mask frame as the indicator. We introduce a binary mask $M \in \{0, 1\}^{1 \times N \times h \times w}$, where 1 indicates the frames to be retained and 0 indicates frames to be generated. While the spatial dimensions of $M$ match those of $z_c$, its temporal length is the same as the target video. Then it is rearranged in a shape of $s \times n \times h \times w$, where $s$ is the temporal span of the encoder. The first frame $I_0$, last frame $I_{N-1}$ and anchor frame $I_a$ are concatenated with zero-padded frames along the time axis to form the guiding frames $I_c \in \mathbb{R}^{C \times N \times H \times W}$. These guiding frames are then compressed into the latent space using the encoder $z_c = E(I_c)$ and $z_c \in \mathbb{R}^{c \times n \times h \times w}$, with $c = 16$ representing the latent channels, $n = 3 + (N - 3)/4^1$, $h = H/8$ and $w = W/8$.

Additionally, we use the image encoder of CLIP Radford et al. (2021) to extract conditioning features $c_0$ and $c_{N-1}$ from the first frame $I_0$ and last frame $I_{N-1}$. These are then concatenated to get a conditioning vector $c_i$:

$$c_i = [c_0, c_{N-1}]. \tag{11}$$

These features are then injected into the DiT Li et al. (2022) model through decoupled cross-attention. The final iterative denoising process is given by

$$z_{t-1} = \text{update}(z_t, u_\theta(z_t; t, m, c_i, c_{P^{\text{fwd}}}, z_c); t), \tag{12}$$

where $m$ represents rearranged masks, $u_\theta(z_t; t, m, c_i, c_{P^{\text{fwd}}}, z_c)$ denotes the output velocity predicted by the model. After a certain denoising process, we obtain a clean latent $z_0$. This latent representation $z_0$ encapsulates the semantic information from the first and last frames, guided by the anchor frame and conditional inputs, and is ready for further decoding to produce the final video.

## 4 Experiments

### 4.1 Implementation Details

**Baseline Models.** To evaluate the effectiveness and generalizability of our method, we integrate it into two state-of-the-art I2V frameworks: Wan 2.1 Wang et al. (2025a) and Hunyuan Video Weijie Kong et al. (2024). For comparison, we include two categories of baseline methods: (1) the current state-of-the-art FLF2V model, Wan2.1-FLF2V Wang et al. (2025a); and (2) video frame interpolation approaches, including ViBiDSampler Yang et al. (2025a) and Generative Inbetweening Wang et al. (2025b).

**Evaluation Metrics.** For evaluation metrics, we use LPIPS Zhang et al. (2018) to evaluate individual frame quality and FVD Unterthiner et al. (2019) for overall video quality. SSIM Wang et al. (2004) and PSNR measure similarity between generated and original videos. Additionally, following StoryEval Wang et al. (2024), we adopt MLLMs, GPT-4o OpenAI (2024) and Gemini 2.0 Flash C Eric Bieber et al. (2025), as additional evaluators to assess the overall video quality from two perspectives: inter-frame consistency and prompt matching degree, with scores up to 100.

---

[1]The constant 3 in this equation represents the total number of control frames after inserting the anchor frame (*i.e.*, the first frame, the anchor frame, and the last frame). In scenarios utilizing multiple anchor frames for FLF2V tasks, this equation generalizes to $n = (N_a + 2) + (N - (N_a + 2))/4$, where $N_a$ is the number of anchor frames.

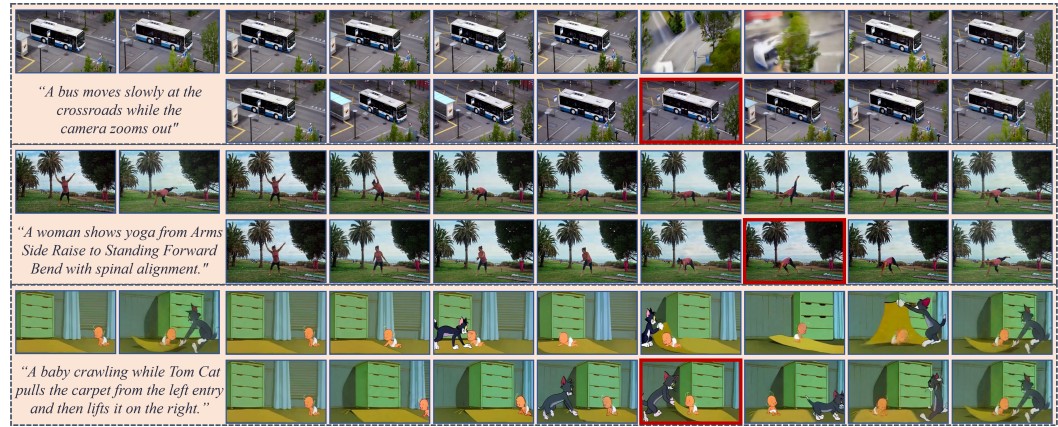

Figure 4: **Qualitative evaluation.** We compare Wan 2.1 and Wan2.1 + AFB across different scenarios. The first row of each group represents the results of Wan 2.1, and the second row represents the results of our method. The anchor frame inserted is highlighted with the red bounding box.

Table 1: Comparison with baseline models.

| Baseline Models | LPIPS↓ | FVD↓ | SSIM↑ | PSNR↑ | GPT-4o ↑ | Gemini 2.0 Flash ↑ |
|---|---|---|---|---|---|---|
| ViBiDSampler | 0.19 | 426.15 | 0.90 | 33.08 | 82.06 | 82.88 |
| Generative inbetweening | 0.24 | 453.76 | 0.85 | 31.25 | 75.42 | 72.15 |
| HunyuanVideo-I2V | 0.25 | 496.32 | 0.82 | 31.48 | 73.28 | 71.69 |
| HunyuanVideo +AFB (Ours) | 0.21 | 435.71 | 0.89 | 32.54 | 81.33 | 79.26 |
| Wan2.1-I2V | 0.22 | 449.68 | 0.87 | 32.13 | 79.31 | 76.43 |
| Wan2.1-FLF2V | 0.19 | 413.68 | 0.91 | 33.20 | 84.23 | 84.94 |
| Wan2.1 +AFB (Ours) | **0.16** | **375.12** | **0.97** | **35.41** | **88.64** | **89.35** |

**Dataset.** At present, there is a notable absence of high-quality, publicly available datasets specifically designed for the FLF2V task. To evaluate the performance of our AFB across different scenarios, we collect a dataset of first and last frames encompassing various scenes such as TV series, indoor/outdoor scene and human poses from the DAVIS Pont-Tuset et al. (2017), the RealEstate10K Salem et al. (2021) and publicly available videos online. We filter the collected videos to create a high-quality dataset featuring significant visual variation between the first and last frames, resulting in a final collection of 436 videos and their corresponding frame pairs. The distribution of our dataset is shown in the Fig.3.

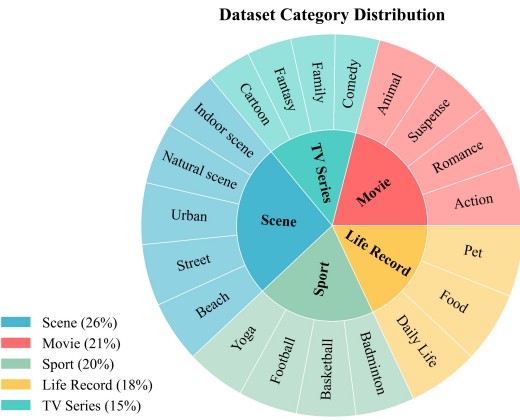

Figure 3: **Distribution of our dataset.**

## 4.2 BASELINE COMPARISONS

**Qualitative Evaluation.** We present a visual comparison between our method and Wan2.1-I2V in Fig.4, showing the superior performance of our approach in terms of motion stability, consistency and overall quality. Under the same input conditions (first-last frames, text prompts), Wan2.1-I2V shows issues in the middle frames of the generated video, such as scene distortion, chaotic human limbs and reduced subject consistency. These issues are effectively mitigated with our AFB. More-

Table 3: **Ablation study on stop step** $K$.

| Stop Step ($K$) | Time (min) | Overhead | FVD ↓ | LPIPS ↓ | SSIM ↑ | PSNR ↑ | GPT-4o↑ | Gemini↑ |
|---|---|---|---|---|---|---|---|---|
| 5 | 23 | +15% | 412.73 | 0.19 | 0.90 | 32.74 | 82.53 | 81.64 |
| **15** | **27** | **+35%** | **388.45** | **0.18** | **0.93** | **33.68** | **85.32** | **86.13** |
| 40 | 37 | +85% | 379.35 | 0.16 | 0.96 | 34.81 | 87.59 | 87.30 |
| 50 | 41 | +105% | 375.12 | 0.16 | 0.97 | 35.41 | 88.64 | 89.35 |

Table 4: Ablation study on text prompts.

| Text prompts | Methods | LPIPS ↓ | FVD ↓ | SSIM ↑ | PSNR ↑ |
|---|---|---|---|---|---|
| General Texts | Wan2.1 | 0.32 | 486.23 | 0.76 | 29.15 |
| General Texts | Wan2.1 + AFB (Ours) | 0.28 | 475.33 | 0.81 | 30.54 |
| Text by Qwen | Wan2.1 | 0.22 | 449.68 | 0.87 | 32.13 |
| Text by Qwen | Wan2.1 + AFB (Ours) | **0.16** | **375.12** | **0.97** | **35.41** |

over, our approach is not only effective in real-world scenarios but also generalizes well to animated scenes, as shown in the third row of Fig.4. More comparison results are available in the Appendix. B.

**Quantitative Evaluation.** We conduct comparative evaluations using our dataset against baseline models. The results are shown in Tab.1. It is evident that our AFB method substantially enhances the overall quality of video generation, along with the smoothness and consistency between the first and last frames, outperforming all original baseline models.

## 4.3 ABLATIONS

**Effect of Multiple Anchor Frames.** We ablate the number of anchor frames ($N_a$) to study how additional constraints affect generation quality. Specifically, we use the quality assessment function $Q$ to identify $N_a$ (where $N_a \in \{1, 2, 3\}$) frames with the lowest quality scores in reverse sequence, and select their mirrored-position counterparts as anchor frames. As shown in Fig. 2, under our

Table 2: Ablation study on multiple anchor frames.

| $N_a$ values | $N_a = 1$ | $N_a = 2$ | $N_a = 3$ |
|---|---|---|---|
| LPIPS ↓ | **0.16** | 0.18 | 0.21 |
| FVD ↓ | **375.12** | 386.94 | 397.50 |
| SSIM ↑ | **0.97** | 0.93 | 0.82 |
| PSNR ↑ | **35.41** | 34.27 | 30.49 |

default video length setting (5 s), using a single anchor frame achieves optimal performance. This finding is further supported by our user study (see Appendix. H for details). We attribute the degradation with more anchors to over-constraining the generation trajectory: multiple anchors may introduce competing constraints, which can reduce motion fluency and diminish generative diversity. In this setting, a single anchor frame is sufficient to mitigate semantic decay while preserving generative diversity, striking an effective balance. For longer videos, however, multi-anchor configurations can be beneficial; see Appendix G for more details.

**Effect of Stop Step** $K$. As described in Sec. 3.2, we terminate the reverse denoising after $K$ steps and perform anchor selection based on the resulting candidate sequence. This stop step $K$ offers a direct efficiency-quality trade-off in AFB[2]. Intuitively, a smaller $K$ reduces computation but provides less denoising and weaker visual cues, whereas a larger $K$ yields more reliable candidates at the cost of higher overhead. As shown in Tab 3,reducing the stop step significantly lowers time overhead. Notably, terminating at timestep 15 ($t = 15$) achieves an optimal balance between performance and cost: the FVD is 388.45, which is very close to the full-pass AFB (375.12) and still significantly outperforms both the original Wan2.1-I2V (449.68) and Wan2.1-FLF2V (413.68), while the inference time increases by only 35% compared to the baseline (27 min vs. 20 min).

---

[2]Except for the ablation in Table 3, all other experiments in the main paper use the full reverse denoising ($K = T$) to maximize generation quality. In practice, $K$ can be adjusted to trade off efficiency and fidelity.

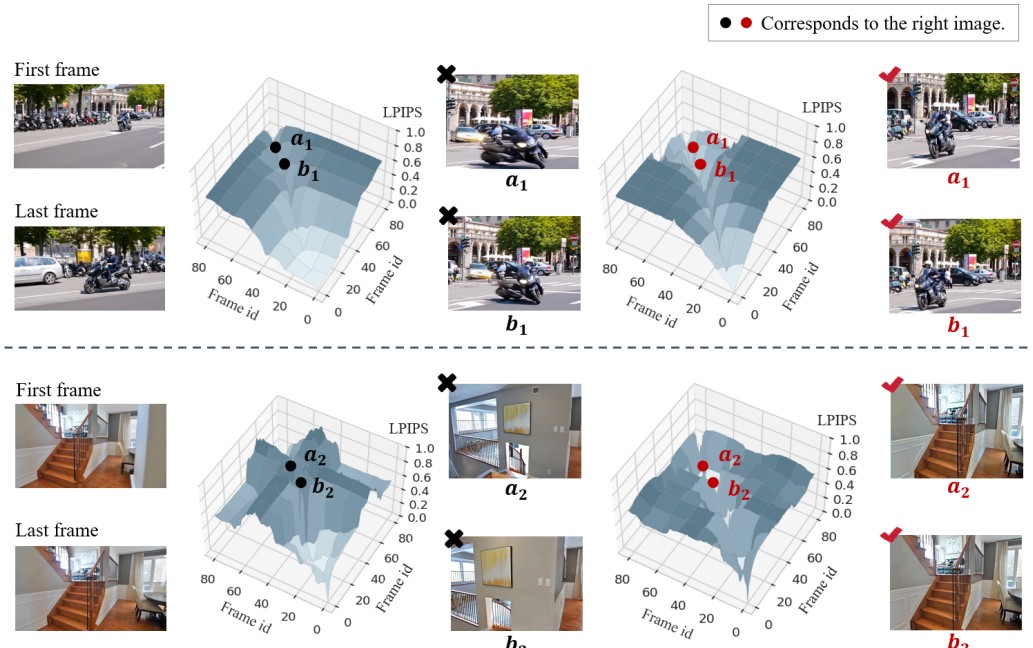

Figure 5: **Inter-frame similarity analysis**. For DAVIS-25 train-69 (Left), the initial frames show almost no change; however, starting from frame 40, a sudden scene shift (motorcycle movement) indicates poor continuity in the intermediate frames. For RealEs-25 95 (Right), the scene switches abruptly to another between frames 40 and 70, demonstrating that the generation of intermediate frames is not controlled by the first and last frames.

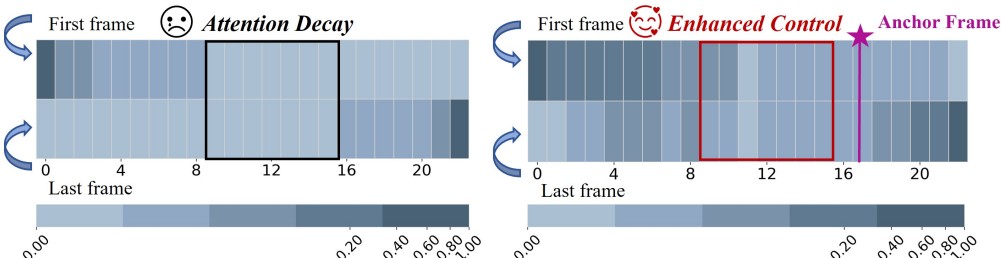

Figure 6: **Visualization of the first and last frames' attention to each intermediate frame.** The left and right figures respectively show the effects before and after the application of our method.

**Effect of Text Prompts.** Tab.4 presents an ablation study on text prompts, comparing specific descriptions generated by Qwen Bai et al. (2023) (customized to align first-last frame semantics) with a general prompt such as "a nice video". The results show that our AFB outperforms Wan2.1-I2V Wang et al. (2025a) in both cases. Moreover, detailed textual guidance enhances the controllability of video generation, meeting the requirements of user-customized generation.

## 4.4 EMPIRICAL STUDY

**Analysis.** To verify whether our proposed AFB effectively bridges the information from the first and last frames to the middle frames, we conduct analytical experiments. We randomly select some samples from our dataset and perform comparative similarity analysis on the video frames generated under first-last frame conditions versus with our AFB method (using the LPIPS metric, where higher values indicate lower similarity). As shown in Fig.5, our AFB successfully bridges the semantics of the first and last frames to the intermediate frames via anchor frames, thus enhancing the overall consistency of the video. In Fig. 6, we visualize and compare the self-attention of the first and last frames to the intermediate frames for our AFB method and the original method. Before incorpo-

rating the anchor frame, the attention maps for the intermediate frames exhibit significant sparsity. After applying our method, this phenomenon is markedly mitigated, which demonstrates that our AFB successfully bridges the semantics of the first and last frames to the intermediate frames via anchor frames, thus enhancing the overall consistency of the video.

## 5 CONCLUSION

First-last frame video generation suffers from semantic degradation in intermediate frames, causing scene distortion and subject deformation that undermine temporal consistency. To address this issue, we propose Anchor Frame Bridging (AFB), a training-free method that mitigates distortion by inserting an anchor frame to bridge the semantics of the first and last frames. Specifically, to obtain an appropriate anchor frame, we introduce an adaptive selection module that dynamically selects the proper anchor frame from candidate sets generated through a reverse process. After that, we develop an anchor frame guided generation approach to generate final video. Experimental results verify that our AFB explicitly enhances the propagation of the first-last frame's semantics, thereby improving the consistency and overall quality of the generated video.

**Discussion.** Our method inherits limitations from the underlying I2V foundation models. While outperforming baseline models in extreme scenarios involving drastic viewpoint changes, severe occlusions and non-rigid deformations, our approach still exhibit motion distortion and physically implausible motion. We analyze these failure cases in the Appendix D.1. Looking forward, we believe these limitations will be mitigated as more powerful video generation models emerge. Moreover, our analysis of semantic information decay and the proposed anchor-frame insertion strategy offers valuable insights for future research on temporal continuity in video generation, particularly for challenging tasks such as long-form generation with distantly spaced start and end frames.

### ACKNOWLEDGMENTS

This work was supported in part by Natural Science Foundation of China (No. U24B6012, 61972217, 32071459, 62176249, 62006133, 62271465, 62406167), the Shenzhen Medical Research Funds in China (No. B2302037), the New Generation Artificial Intelligence-National Science and Technology Major Project (No. 2025ZD0122702), and AI for Science (AI4S)-Preferred Program, Peking University Shenzhen Graduate School, China.

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

## A  THE USE OF LLMs

In the preparation of this manuscript, we leveraged Large Language Models (LLMs) to refine the text for improved clarity. Furthermore, we use LLM to generate image descriptions which served as one of the control condition for video generation process, as detailed in Sec. 3, and to provide auxiliary evaluation metric for our experimental results as detailed in Sec. 4.1.

## B  MORE VISUALIZATION RESULTS

The current FLF2V (First-Last Frame to Video) model exhibits several limitations when generating videos from given start and end frames, as shown in Fig. 7. These issues include unnatural and abrupt transitions—particularly toward the end of the sequence—where the video appears to forcibly align with the provided last frame. Additionally, the model produces implausible limb distortions of the main character, which become especially noticeable during high-intensity motions. Other artifacts, such as ghosting, further degrade the visual quality and coherence of the generated videos.

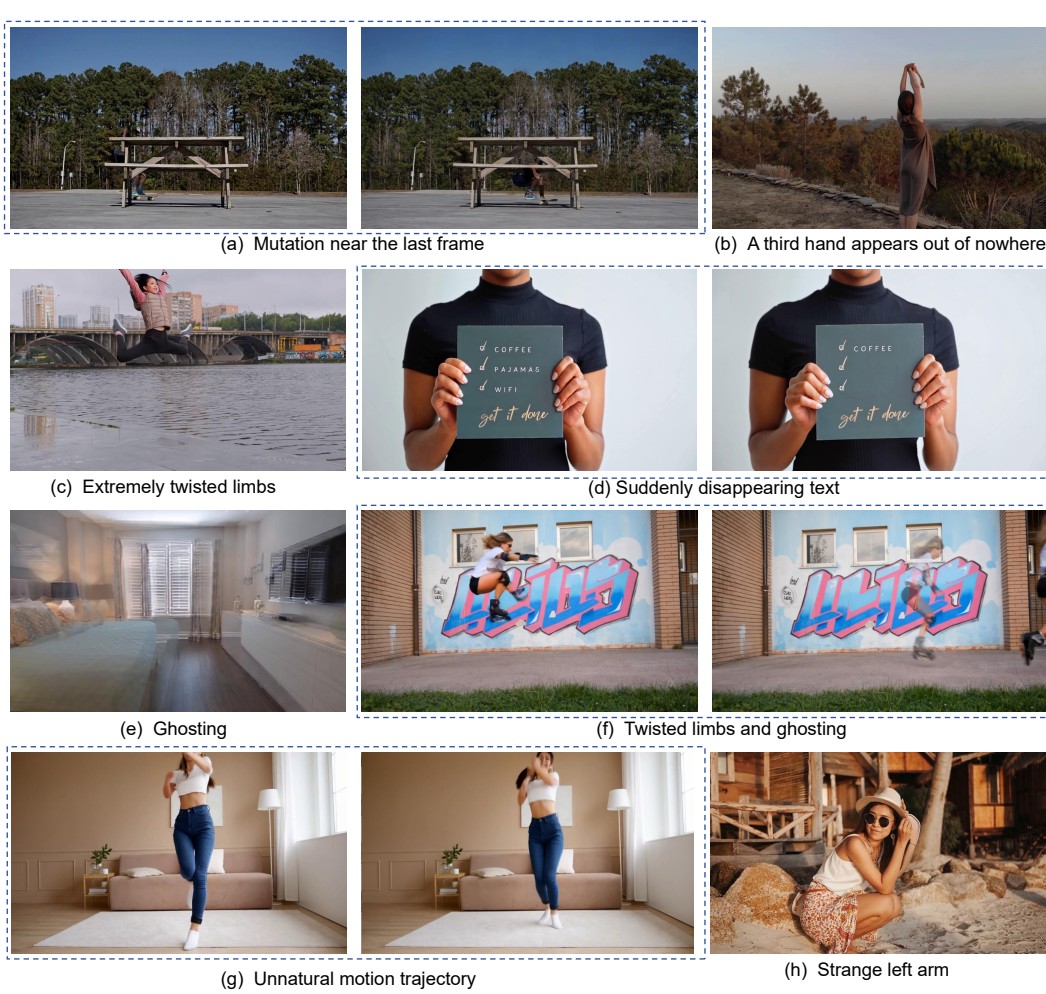

(a) Mutation near the last frame    (b) A third hand appears out of nowhere

(c) Extremely twisted limbs    (d) Suddenly disappearing text

(e) Ghosting    (f) Twisted limbs and ghosting

(g) Unnatural motion trajectory    (h) Strange left arm

Figure 7: **Disadvantages of generating videos.**

The detailed experimental results are presented in Fig. 8. Each unit displays the provided first and last frames along with the corresponding text prompts on the left. On the right, the top and bottom rows show the video sequences generated without and with our method, respectively. A comparison between the two rows clearly demonstrates that our AFB module substantially enhances the alignment with the given start and end frames. For instance, in the third example, distorted body movements present in the original output are effectively corrected after introducing the anchor

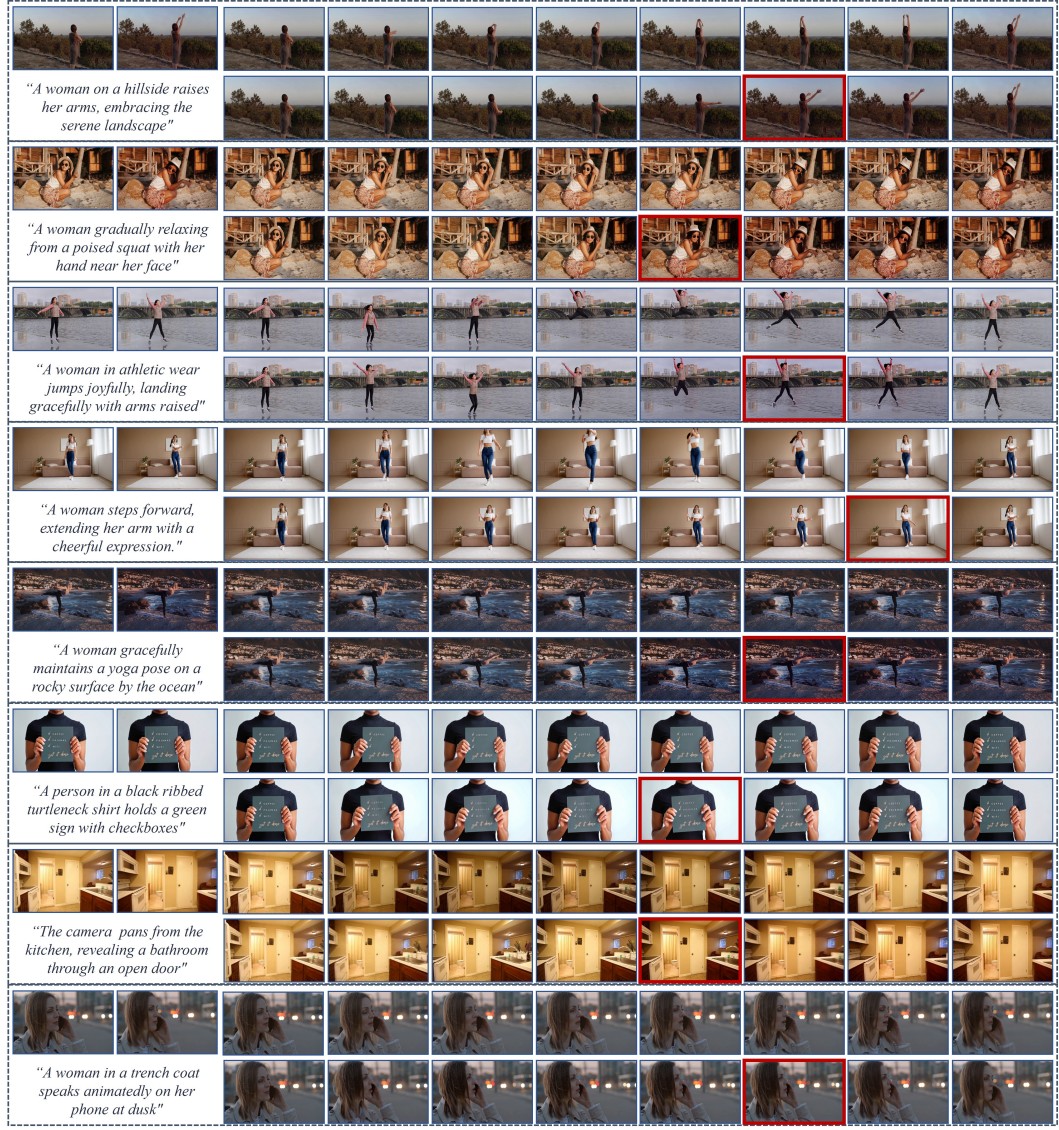

Figure 8: **More Visualization Results.** In each unit, the top row shows the results generated by original FLF2V model, and the bottom row shows the videos generated after applying our AFB method. The added anchor frame are highlighted with a red box.

frame, leading to more natural and coherent motion. Similarly, in the seventh case, the original video exhibits an unnatural phenomenon where water flows from the faucet and stops abruptly without external interaction—an artifact that is successfully resolved with our approach.

## C   COMPARISON OF DIFFERENT ANCHOR FRAME INSERTION STRATEGIES

In the Anchor Frame Guided Generation step, we conducted a qualitative ablation study comparing two anchor frame insertion strategies. For Wan 2.1 Wang et al. (2025a), the first approach averages the latent representation of the anchor frame, $z_c^f$, with the noise latent, $z_t^f$, at each denoising step, where $f$ indicates the anchor frame position. In contrast, the second approach follows the original Wan 2.1 setup, applying no additional averaging operation. The comparison results are presented in Fig. 9. The observations indicate that the second approach outperforms the first, as the first method leads to severe grayish blurring in the intermediate frames and fails to produce plausible video sequences. Therefore, the second approach is adopted as the final method.

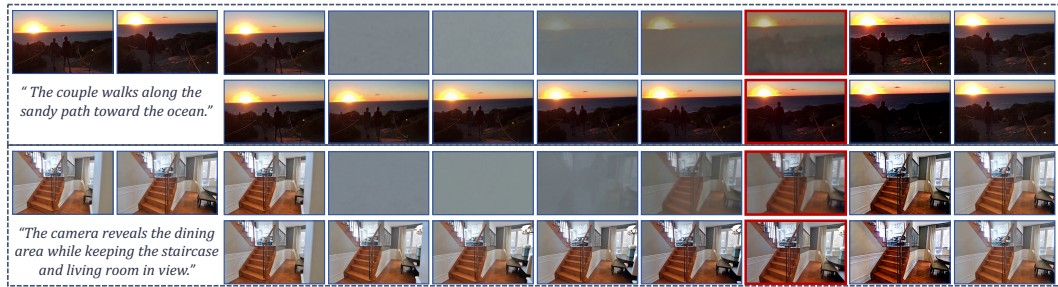

Figure 9: **Qualitative comparison of anchor frame insertion strategies.** In each unit, the top row illustrates the anchor-frame averaging approach, and the bottom row shows the direct conditioning approach without averaging. The added anchor frames are highlighted with a red box.

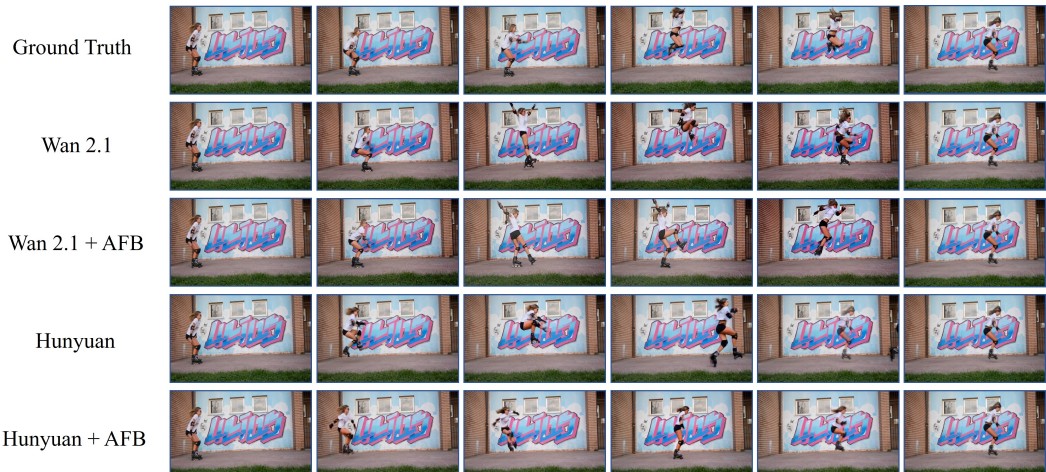

Figure 10: **Complex scenarios.** Existing I2V models struggle with complex scenarios, such as those featuring non-rigid motion, extreme viewpoint changes, or severe occlusions. Although our method yields observable improvements, it does not fully resolve inherent issues like joint distortion and motion blur, which remain challenging.

# D  MORE DISCUSSION

## D.1  CHALLENGING TASKS

Our method is limited by the capabilities of the baseline I2V model. For challenging tasks invovling non-rigid motion, severe occlusions and Drastic viewpoint shifts, It is difficult for existing I2V models to accurately predict object trajectories and human body movements. As illustrated in the Fig. 10, while the insertion of our anchor frame yields some improvement over the baseline model, the motion sequence remains unnatural when compared to the ground truth. Artifacts such as physically implausible distortions and ghosting effects are still present. However, we believe that with the ongoing advancements in foundation models, combined with our method, these challenging issues will be gradually resolved in the near future.

## D.2  RESEARCH ON PHYSICAL CAUSALITY

Our AFB method also demonstrates effectiveness in scenarios with strong physical causality (e.g., an orange falling into water, glass shattering, water droplet ripples, smoke dissipation, etc.). This is because the efficacy of AFB does not rely on a perfectly coherent reversed video—the reverse generation process merely serves as a means to obtain high-quality anchor frames that are highly relevant to the first and last frames. As long as we can obtain an anchor frame that conforms to physical laws and maintains strong relevance to the first and last frames, AFB can function effectively, and this can be achieved through the reverse process.

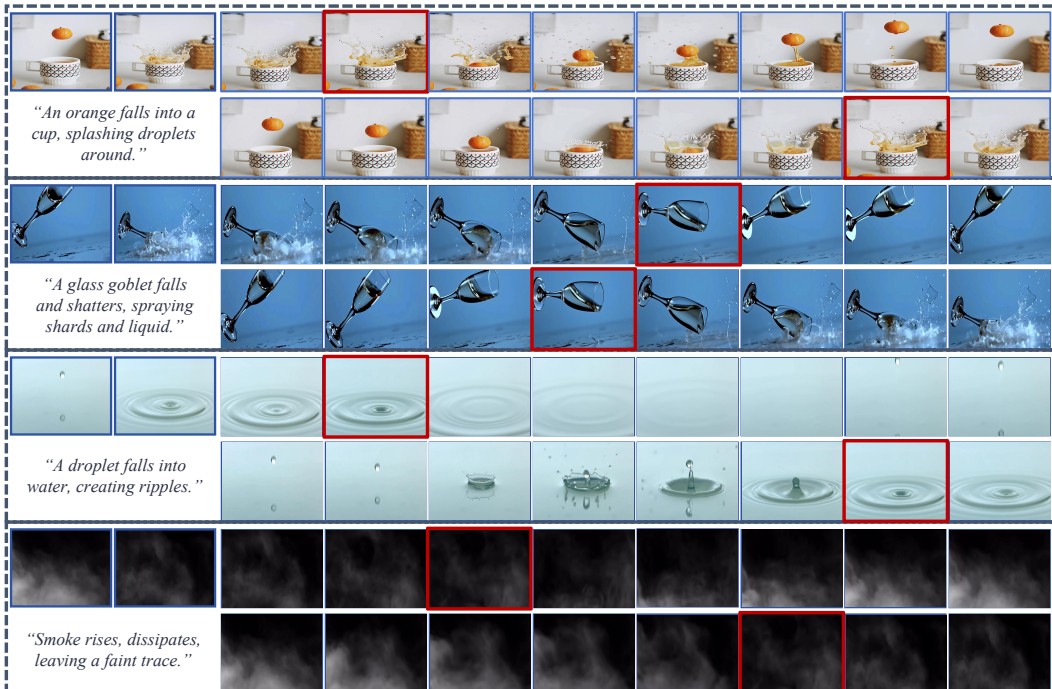

Figure 11: **Application examples of our method in scenarios with strong physical causality.** Row 1: Video generated by reversing the first and last frames, where red boxes indicate the anchor frames selected during the reversal process; Row 2: Results using our method (AFB) with forward generation between the first and last frames, where red boxes represent inserted anchor frames.

Experimental results indicate that video generation based solely on first and last frames often produces poor results near the last frame, leading to continuity breaks. Therefore, after mirroring the sequence, we select frames from the early segment of the reversed video as anchor frames. As shown in Figure 11, although the full reversed video exhibits physical distortions in its later part, the initial segment remains highly faithful. This suggests that even in extreme scenarios, the early stage of reverse generation still serves as a highly reliable semantic source, enabling AFB to extract anchor frames from this interval and guide coherent forward video generation.

### D.3 ANALYSIS OF ASYMMETRIC MOTION

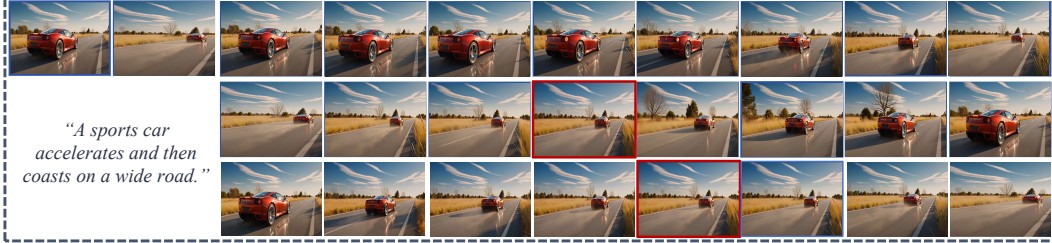

Figure 12: **Cases of Asymmetric Motion.** Row 1 depicts the results generated using only the first and last frames; Row 2 shows the output from reversing the first and last frames, with the selected anchor frame highlighted by a red box; Row 3 presents the generation after inserting the anchor frame, where the inserted anchor frame is marked with a red box.

To investigate the impact of asymmetric motion on our method, we test the "car accelerating and then coasting" example. As illustrated in Figure 12, the decay of semantic constraints manifests similarly in both forward and reverse generations, regardless of whether the car is fast or slow. Specifically, both directions exhibit motion discontinuities, such as the car failing to initiate acceleration naturally.

In contrast, after applying AFB, the anchor frame effectively bridges this semantic gap, rendering the acceleration process significantly smoother and physically coherent.

### D.4  ANALYSIS OF COMPLEXITY DIFFERENCE BETWEEN FIRST AND LAST FRAMES

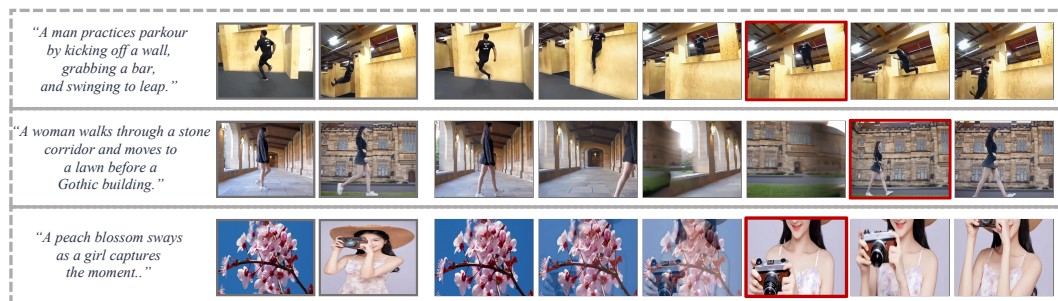

Figure 13: **Analysis of Complexity Difference Between First and Last Frames.** The red box indicates the inserted anchor frame.

We conduct evaluations on three specific scenarios representing different challenges (Figure 13). Our findings are as follows: **Complex Motion** (e.g., Parkour): Despite the highly dynamic nature of the motion, our method successfully identifies a valid anchor frame, generating coherent high-speed actions. **High Complexity Variance with Continuity** (e.g., Character Scene Transition): In cases where the starting and ending backgrounds differ significantly but the subject remains consistent (e.g., the same character walking from a corridor to the outdoors), AFB still functions effectively. The semantic continuity of the subject allows the anchor frame to bridge the complexity gap. **High Complexity Variance without Continuity** (e.g., Flower → Girl): In extreme cases where the first and last frames are semantically unrelated, .the generated result resembles a disjointed, slideshow-like transition rather than a coherent video. However, we wish to clarify that the primary objective of AFB is to preserve physical and semantic continuity; therefore, semantically unrelated boundary frames fall outside the scope of our current method. While AFB is not designed to resolve this semantic disjointedness, addressing such non-continuous constraints remains an interesting direction for future work.

## E  VALIDATION OF CONTINUITY BREAKPOINTS BETWEEN FORWARD AND REVERSE GENERATION

To validate the robustness of the hypothesis that *continuity breakpoints* occur at consistent positions in both forward and reverse generation, we conduct large-scale quantitative experiments on a complete dataset encompassing diverse motion types (e.g., asymmetric motion, severe deformation). We computed frame-wise average LPIPS curves for all 436 videos during forward generation $(I_0 \rightarrow I_{N-1})$ and reverse generation $(I_{N-1} \rightarrow I_0)$. Statistical result (Figure 14) reveals that: the quality collapse point (LPIPS peak) in forward generation occurs at frame 56, while in reverse generation it appears at frame 55, with merely a 1-frame absolute positional deviation.

This provides strong statistical evidence that, regardless of video content, generation direction, or motion complexity, quality collapse points tend to occur at statistically consistent relative positions. This symmetry is not coincidental but fundamentally driven by "information decay"—a systematic issue dominated by model architecture and error accumulation, largely independent of generation direction.

Specifically, most FLF2V models are adapted from I2V architectures. Their inherent characteristic is that the semantic constraints from the first and last frames gradually weaken over temporal progression (as shown in Figure 1 (a)). During forward generation, as frames move farther from the first frame, diminishing semantic constraints allow minor visual artifacts to remain uncorrected. These subtle deviations propagate, accumulate, and amplify frame-by-frame (Error Accumulation), eventually causing the generated content to diverge from plausible trajectories and triggering continuity breaks at specific "semantic vacuum zones" (as shown in Fig. analysis).

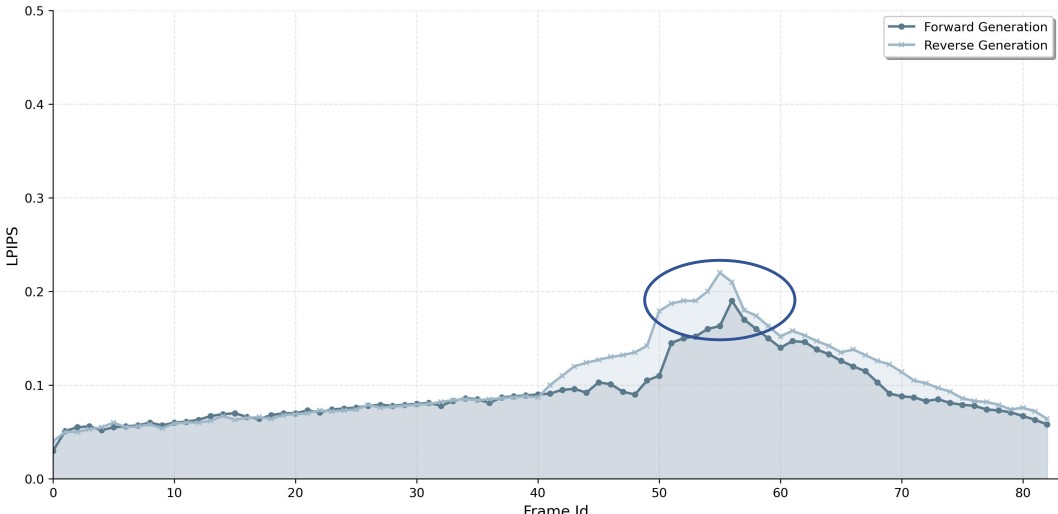

Figure 14: **Average LPIPS between adjacent frames in forward and reverse generation.** Blue circles indicate positions where LPIPS exhibits extremum values, representing continuity breaks.

Although reverse generation alters the direction, the mechanisms of "constraint decay" and "error accumulation" persist, leading to continuity breaks at positions similar to those in the forward process. AFB strategically leverages reverse generation to locate the most severe breakpoints and inserts high-quality anchor frames at corresponding positions in the forward process, thereby mitigating information decay at these critical locations.

## F  ADDITIONAL BASELINE COMPARISON

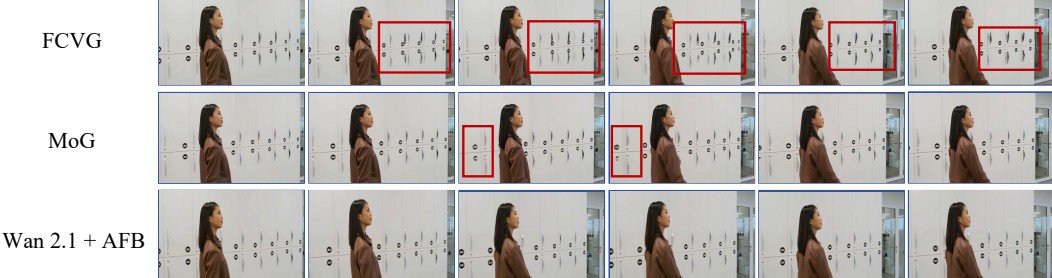

Figure 15: **Qualitative evaluation.** Red box highlights the defective portions in generated videos.

Table 5: **Quantitative evaluation.** The best results are **bolded**.

| Baseline Models | LPIPS↓ | FVD↓ | SSIM↑ | PSNR↑ |
|---|---|---|---|---|
| FCVG | 0.19 | 438.83 | 0.86 | 33.62 |
| MoG | 0.26 | 421.53 | 0.69 | 18.41 |
| Wan2.1 +AFB (Ours) | **0.16** | **375.12** | **0.97** | **35.41** |

In Tab. 1, we compare against recent baselines under our default setting of generating 5-second videos. As FCVG Zhu et al. (2025) and MoG Zhang et al. (2025) are typically evaluated on shorter clips (approximately 1–2 seconds), we include their comparisons here in the supplementary material. Qualitative results are shown in Figure 15, and quantitative results are presented in Table 5. FCVG and MoG typically generate short clips (approx. 1-2 seconds). In contrast, AFB generates longer, complex videos (approx. 5 seconds). As shown in Table 5, AFB significantly outperforms both FCVG and MoG. Specifically, AFB achieves the lowest FVD (375.12) and the highest PSNR

(35.41), demonstrating superior temporal coherence and visual fidelity. As detailed in Figure 15, Both FCVG and MoG exhibit noticeable ghosting artifacts on static objects, such as the cabinet handles (highlighted in red boxes). In contrast, AFB leverages semantic anchors to generate clean, artifact-free frames with better logical coherence.

## G  ANALYSIS OF LONG VIDEOS WITH MULTIPLE ANCHOR FRAMES

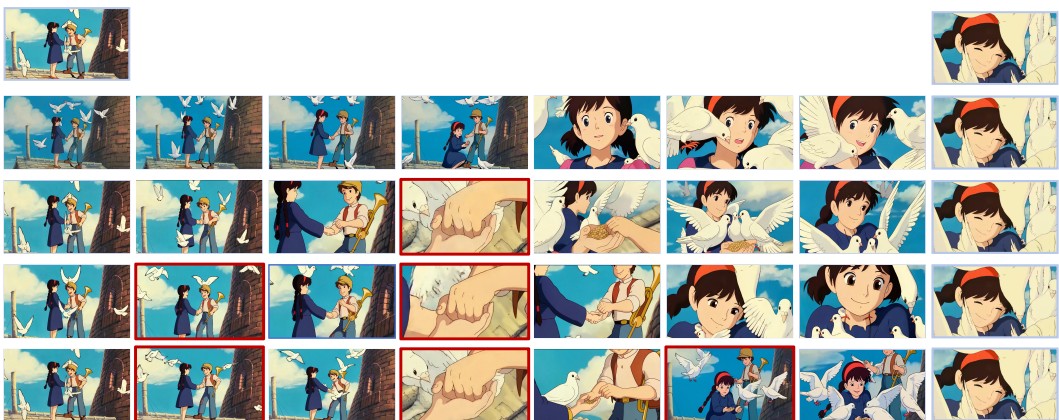

*"A girl and a boy meet and talk on the rooftop. He gives her bird feed, and as she offers it, a pigeon lands on her hands to eat."*

Figure 16: **Long videos with multiple anchor frames.** Row 1 shows the generation using only the first and last frames. Row 2 presents the result with one anchor frame inserted. Row 3 displays the output with two anchor frames added. Row 4 illustrates the generation with three anchor frames incorporated. All inserted anchor frames are highlighted by red boxes.

For further analysis, we employ Jimeng AI to generate 10-second long videos. The results (Figure 16) demonstrate that for long-video generation, a multi-anchor strategy (inserting 2 or 3 frames) yields superior performance compared to the single-anchor approach. As the number of anchors increases, the temporal coherence of the intermediate frames also improves. For future work on longer videos and more complex scenarios, we plan to extend AFB by exploring adaptive multi-anchor insertion methods.

## H  USER STUDY

We design a questionnaire and distribute it to different participants. In the questionnaire, we compare the results of Wan 2.1 and Wan 2.1 + AFB in three aspects: video consistency, text alignment and generated video quality. We also evaluated how the number of anchor frames, $K$, affects generation quality, assessing our method across $K \in \{1, 2, 3\}$. Further details of the questionnaire are presented below. A total of 52 valid responses are collected and analyzed, with the statistical results shown in Fig.17. It can be observed that our method is more favored by users compared to using Wan 2.1 alone. Additionally, among the different numbers of anchor frames tested, users clearly prefer the configuration with a single anchor frame ($K = 1$).

Our Google Form survey consists of 4 sections, with a total of 33 questions. In the first three sections, we compare our AFB with Wan2.1 Wang et al. (2025a). In the final section, we compare videos generated with different frame insertion positions in AFB. Specific information and screenshots of the questionnaire are as follows:

1. **Consistency in video generation.** Users are asked to evaluate the temporal consistency of the generated videos with the given first and last frames. For instance, when provided with two images of a girl raising her hands alongside a text description, users are instructed to select the video demonstrating higher consistency. See Fig. 18 for details.

2. **Text Alignment in Video Generation.** Users assess whether the video content accurately reflect the provided text descriptions. For instance, given a text prompt describing a smooth

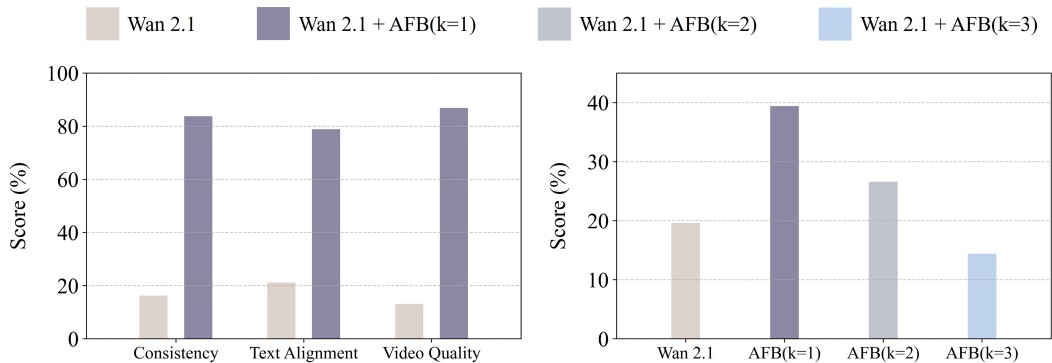

Figure 17: **User Study**. The left figure shows user evaluations of Wan 2.1 Wang et al. (2025a) and Wan 2.1 + AFB in terms of video consistency, text alignment and generated video quality. The right figure shows user evaluations on how number of anchor frames impacts video generation quality.

> camera transition from a kitchen to a bathroom along with first and last frames, users are instructed to select the video that best matched the text prompt. See Fig. 19.

3. **Visual Quality of Video Generation.** Users select the video that exhibit the highest overall visual quality. For example, given a prompt such as "A woman gracefully maintains a yoga pose on a rocky surface by the ocean." along with first and last frames, users choose the video displaying superior visual fidelity, such as realistic water ripples or detailed rock textures. See Fig. 20.

4. **Impact of Multiple Anchor Frames.** Users compare videos generated using $K$ anchor frames ($K \in 1, 2, 3$) alongside the original Wan 2.1 FLF2V output. They are instructed to vote for the video with the best overall quality in each case. The interface for this task is shown in Fig. 21.

In the user study, participants evaluate nine groups of videos in the first three sections and six in the final section. For each group, they select the video they consider to be of the highest quality. Each selection counts as one point. We collect a total of 52 valid responses, and the aggregated results are visualized in Fig. 17.

**Consistency in video generation**

In this section, you will be asked to select the video that you believe best aligns with the given first and last frames and has good continuity. Please focus on the coherence of the video and how well it matches the information conveyed by first-last frames. Note that video quality is not a factor in this evaluation.

The conditions for the generated video are provided, including the first-last frames as well as text descriptions.

You only need to select the one video you believe is the best; there is no need for multiple selections or ranking.

Which video has the best consistency? *

first frame                    last frame

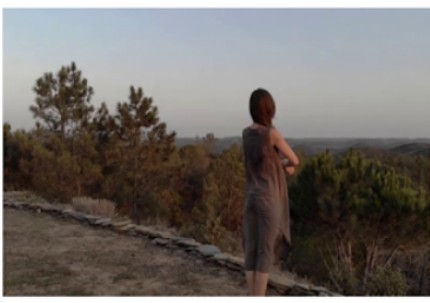 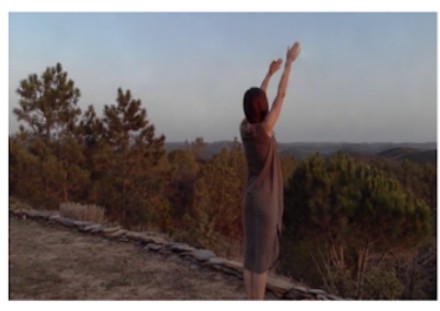

*"A woman standing on a hillside with her back to the camera gradually raises her arms above her head, embracing the serene landscape of trees and rolling hills under a soft sky."*

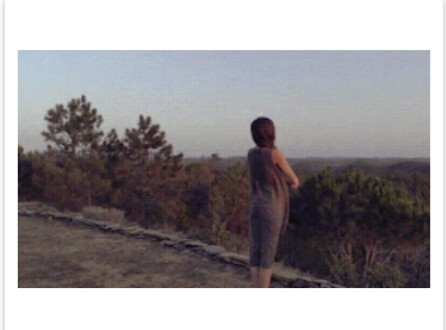 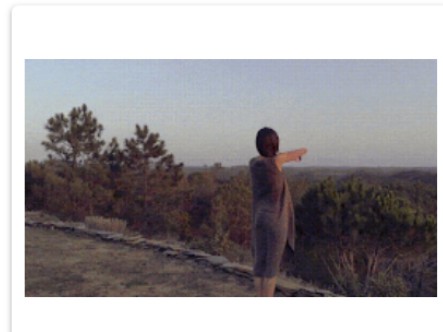

○ a                    ○ b

Figure 18: **The instruction and example for user study.**

**Text alignment in video generation**

In this section, you will be asked to select the video that you believe best aligns with the given text descriptions. Please also take the quality of the video or frames into consideration. For instance, videos with distorted limbs or deformed movements will be considered less favorable.

The conditions for the generated video are provided, including the first-last frames as well as text descriptions.

You only need to select the one video you believe is the best; there is no need for multiple selections or ranking.

Which video has the best alignment with the given text descirption? *

first frame                                      last frame

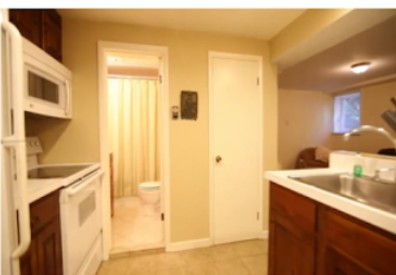          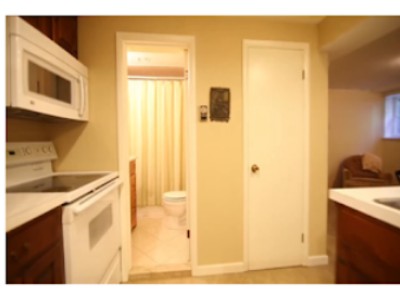

*"The camera smoothly transitions from a wide view of a cozy kitchen and bathroom area to a closer, more focused shot of the same space, gradually removing the sink and countertop from the frame while maintaining the visibility of the stove, microwave, and open doorway leading to the bathroom."*

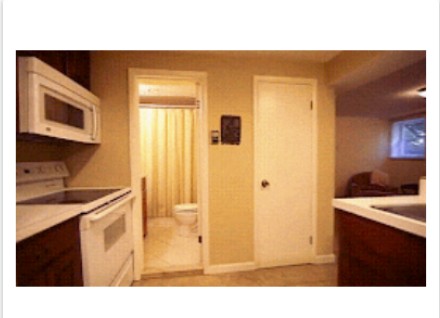          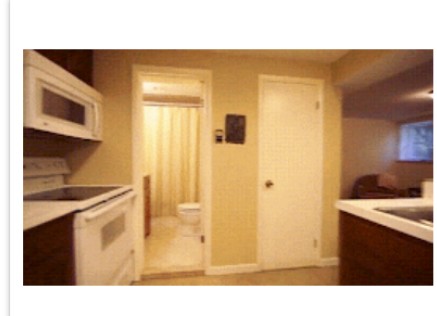

○  a                                              ○  b

Figure 19: **The instruction and example for user study.**

**Quality in video generation**

In this section, you will be asked to select the videos with the best visual quality.
The conditions for the generated video are provided, including the first-last frames as well as text descriptions.
You only need to select the one video you believe is the best; there is no need for multiple selections or ranking.

Which video has the best visual quality? *

first frame

last frame

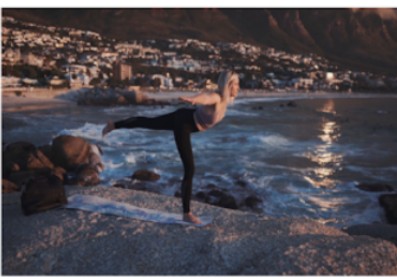 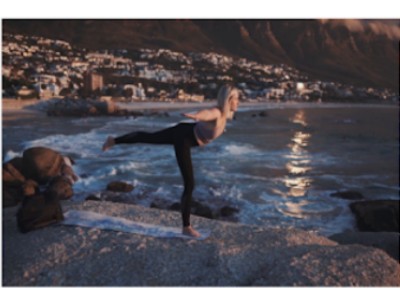

*"A woman gracefully maintains a yoga pose on a rocky surface by the ocean, transitioning smoothly from one stance to another while the waves and coastal town remain a serene backdrop."*

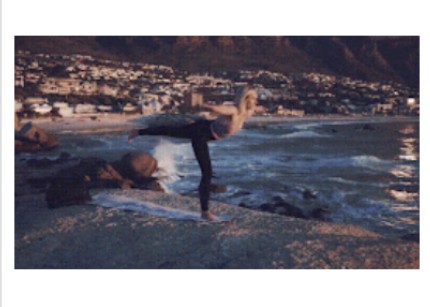 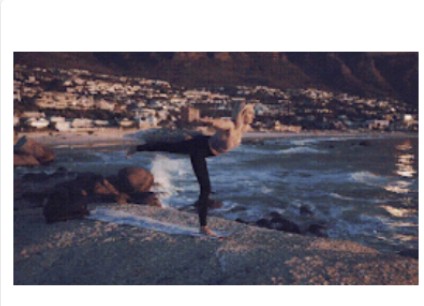

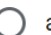 a          ○ b

Figure 20: **The instruction and example for user study.**

**Evaluation of Video Generation Methods**

In this section, you will be asked to evaluate videos that differ due to slight variations in generation methods. Please select the video you prefer based on overall quality and coherence.
The conditions for the generated video are provided, including the first-last frames as well as text descriptions.
You only need to select the one video you believe is the best; there is no need for multiple selections or ranking.

Which video you believe is the best one? *

first frame                                    last frame

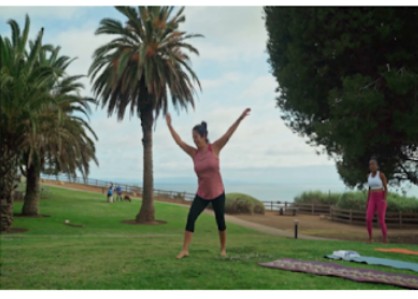  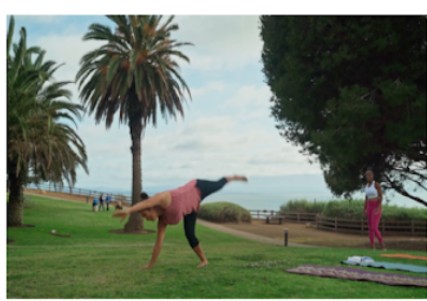

*"A woman gracefully transitions from a standing pose with arms outstretched to a dynamic handstand on a grassy park area, while another person observes and the serene backdrop of palm trees and the ocean remains constant."*

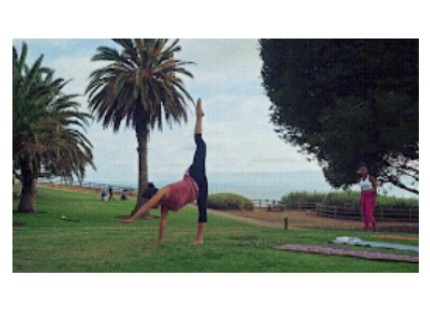  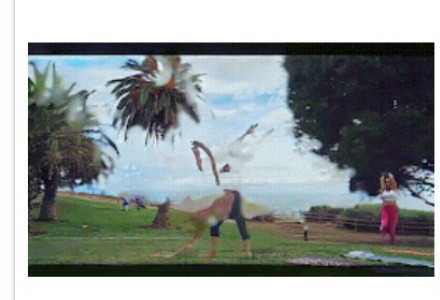

○ a                                    ○ b

Figure 21: **The instruction and example for user study.**

