# OpenReview forum: "Anchor Frame Bridging for Coherent First-Last Frame Video Generation"
_ICLR.cc/2026/Conference — ICLR 2026 Poster_

### Official Review · Reviewer_hmuv · 2025-10-30

**Soundness:** 3
**Presentation:** 2
**Contribution:** 2
**Rating:** 6
**Confidence:** 4

**Summary:**

This paper investigates the challenge of semantic degradation and temporal inconsistency in first-last frame video generation using diffusion models. The authors propose the Anchor Frame Bridging (AFB) method, which automatically selects and inserts an adaptive anchor frame at points of maximal semantic discontinuity in the intermediate video. Their approach consists of an adaptive selection mechanism using LPIPS-guided frame evaluation and a guided generation process augmenting the standard pipeline with anchor frames. The method is designed as a plug-and-play module, requiring no additional training and is validated on newly curated benchmarks and multiple image-to-video diffusion backbones.

**Strengths:**

- **Clear Targeted Problem**: The paper identifies and clearly explains a critical shortcoming in current first/last-frame-to-video models—semantic drift and abrupt transitions in intermediate frames (see intro, Fig. 1).
- **Methodological Innovation**: AFB uses a reverse-generation heuristic and LPIPS-based metric to localize semantic discontinuities and insert an anchor frame, effectively bridging boundary semantics throughout the sequence (Method Sec. 3, Eqns p4–p5, Fig. 2).
- **Empirical Rigor**: The authors provide both qualitative and quantitative benchmarks, comparing AFB-augmented models against strong baselines (including Wan2.1, Hunyuan Video, ViBiDSampler, Generative Inbetweening) on a purpose-built test set.
- **Comprehensive Ablation**: Multiple dimensions (number/position of anchors, prompt variants, effect of diffusion steps) are dissected with quantitative evidence (Tables 1, 2, Fig. 5a).

**Weaknesses:**

- **Limited Theoretical Insight into Anchor Utility**:

While the method for selecting anchor frames is well-motivated empirically, there is only an informal rationale for why mirroring the LPIPS-based breakpoint in a reverse diffusion pass is optimal. The mathematical treatment (Sec. 3.2, Eqns p5) relies on an empirical proxy rather than theoretical guarantees. This leaves open whether the anchor truly maximizes semantic continuity for all types of motion or scenes, or merely correlates empirically. A more thorough theoretical foundation is desirable to formally ground the anchor selection beyond heuristic evidence.

- **Complexity in Hyperparameter Selection and Heuristic Choices**:

The anchor frame is chosen based on the peak of a smoothed local LPIPS curve at a fixed (empirically chosen) denoising timestep (Fig. 5b). But the paper provides limited discussion of sensitivity to these hyperparameters (e.g., LPIPS window size, selected denoising phase, anchor mirroring formula). There is a risk that such choices may not generalize, or that their necessity complicates adoption.

- **Visual Quality Trade-offs in Extreme Cases**:

While the reported objective and subjective gains are strong (Table 1), failure cases and remaining artifacts (Appendix D, Fig. 11) suggest that AFB does not fully resolve physical implausibility or distortions in highly dynamic settings. This is acknowledged, but could be further emphasized as a hard limitation.

**Questions:**

- Could the authors clarify if any formal guarantees or theoretical results support the observed empirical optimality of the mirrored LPIPS-breakpoint approach over alternative anchor selection strategies or more adaptive criterion? What failure cases or counterexamples have you observed?

- Can you elaborate on the impact and selection process for the denoising timestep at which LPIPS is computed, the window width for local averaging, and any anchor selection heuristics?

---

> ### Author Response · Authors · 2025-11-24
> **Response to Reviewer hmuv**
>
> We sincerely thank Reviewer for the valuable comments. We are encouraged that the reviewer recognizes our **“methodological innovation”** in utilizing the reverse-generation heuristic to **“effectively bridge boundary semantics”** throughout the sequence. We also appreciate the acknowledgement of our **“empirical rigor”** in benchmarking against strong baselines and the **“comprehensive ablation”** studies supported by **“quantitative evidence.”** In the following, we provide detailed responses to the weaknesses and questions raised. We hope our response fully resolves your concerns.
>
> > **Response to Weakness 1 (Limited Theoretical Insight):**
>
> Thank you for your valuable questions. Firstly, we wish to clarify that while our strategy is heuristic, it is not based on coincidence but is deeply rooted in the intrinsic **"information attenuation"** mechanism of FLF2V models. This attenuation is a systematic issue driven by model architecture and error accumulation, which is largely independent of the generation direction.
>
> 1. **Mechanism of Information Attenuation**: Most FLF2V models are adapted from I2V architectures. Their inherent characteristic determines that the semantic constraints from boundary frames gradually decay over time (as analyzed in  Fig.1(a) of the main text). During forward generation, as the distance from the first frame increases, the semantic constraint weakens, causing minor visual flaws in each frame to go uncorrected. These tiny deviations **propagate, accumulate, and amplify (Error Accumulation)** frame-by-frame, eventually causing the generated content to deviate from a plausible trajectory and triggering a continuity break at a specific "semantic vacuum" .  Although reverse generation changes direction, the mechanisms of "constraint decay" and "error accumulation" persist, inevitably leading to a continuity break at a **similar position**.
>
> 2. **Quantitative Verification:** To verify the robustness of this assumption, we conduct a quantitative experiment across our dataset, encompassing diverse motion types such as asymmetric motion and drastic deformation. We calculate the frame-wise average LPIPS scores for both forward generation ($I _ 0 \to I _ {N-1}$) and reverse generation ($I _ {N-1} \to I _ 0$) across all videos. The statistical results (**summarized in the table below and detailed in Appendix G Figure 18**) reveal that the quality breakdown point (peak LPIPS) for forward generation occurs at Frame 56, while for reverse generation, it occurs at Frame 55. The absolute deviation is merely 1 frame. This provides strong statistical evidence that the "positional consistency" between forward and reverse generation is robust. Regardless of the video content and motion complexity, the point of quality collapse in the reverse generation consistently aligns with that of the forward generation.
>
> **Table: Comparison of average LPIPS scores (lower is better) between Forward ($I_0 \to I_{N-1}$) and Reverse ($I_{N-1} \to I_0$) generation across 436 videos.**
>
> | Frame ID | 0 | 5 | 10 | 15 | 20 | 25 | 30 | 35 | 40 | 45 | 50 | **55** | 60 | 65 | 70 | 75 | 80 |
> | :--- | :---: | :---: | :---: | :---: | :---: | :---: | :---: | :---: | :---: | :---: | :---: | :---: | :---: | :---: | :---: | :---: | :---: |
> | Forward | 0.03 | 0.055 | 0.06 | 0.07 | 0.07 | 0.076 | 0.08 | 0.085 | 0.09 | 0.103 | 0.11 | **0.163** | 0.14 | 0.126 | 0.088 | 0.079 | 0.067 |
> | Reverse | 0.04 | 0.06 | 0.059 | 0.063 | 0.069 | 0.074 | 0.079 | 0.084 | 0.087 | 0.127 | 0.179 | **0.22** | 0.152 | 0.152 | 0.135 | 0.114 | 0.086 |
>
> 3. Theoretical Perspective：We agree with your suggestion to provide deeper insight.  Specifically, we can conceptualize video generation as a **sequential decision process**. Each generation step introduces a microscopic error $\epsilon$. Simultaneously, the **Mutual Information** with the condition frame (e.g., $I\_0$) decays over temporal distance $t$, which can be modeled as a decay function $D(t)$.Since the forward and reverse processes share the exact same model architecture and parameters, they are governed by the same underlying mechanism of information attenuation.
>
>  While establishing a rigorous mathematical bound covering all scenarios is mathematically intractable due to the high-dimensional complexity and instance-level variance of video data,  our experimental results strongly confirm that the **systematic bias** (architecture-driven decay) dominates over instance-level noise. This validates the robustness of our assumption. In the future, we also plan to explore Stochastic Differential Equations (SDEs) as a potential framework to mathematically model this decay process, trying to provide deeper theoretical insights.

---

> ### Author Response · Authors · 2025-11-24
> **Response to Reviewer hmuv**
>
> > **Response to Weakness 2 (**Hyperparameter Selection and Heuristic Choices**):**
>
> Thank you for the **detailed inquiry regarding hyperparameter sensitivity.** Firstly, we wish to clarify that the parameters in AFB are not arbitrary heuristics but are derived from careful consideration of model characteristics. We have conducted various experiments to demonstrate their robustness.
>
> 1. **LPIPS Window Size:** As mentioned in the paper, LPIPS utilizes pre-trained deep neural network features to simulate human visual perception. As a perceptual metric, it focuses on visual and structural differences, making it inherently robust to high-frequency pixel noise. A "semantic break" in video generation typically manifests as an abrupt change in coherence between adjacent frames. Using the **minimal window size** (calculating only adjacent frames, $Q(I\_n) = - \frac{1}{2}(\text{LPIPS}(I\_{n-1}, I\_n) + \text{LPIPS}(I\_n, I\_{n+1}))$) is sufficient to keenly capture these continuity changes and precisely localize the breakdown point $\alpha$. Increasing the window size would introduce unnecessary computational overhead and necessitate averaging the coherence changes over a larger range, which risks smoothing out the critical local variations needed for precise localization.
>
> 2. **Selected Denoising Phase:** Regarding the choice of denoising phase, our original choice was timestep 45 (t=45). As discussed in the main text, assessing frame quality is critical for anchor localization. As shown in Fig 5.(b), in early sampling stages, insufficient denoising results in ambiguous image features, undermining the reliability of LPIPS scores for anchor selection.
>
> However, to address efficiency concerns, we explored robust anchor selection at earlier stages and conducted supplementary experiments. We propose the Fast-AFB strategy (see Appendix L), which shifts the evaluation target from the noisy latent $x_t$ to the predicted clean frame $\hat{x}_0$ derived from the current step.
>
>    1.) **Results:** As shown in the table below, reducing the stop step significantly lowers time overhead. **Notably, terminating at timestep 15 ($t=15$) achieves an optimal balance between performance and cost:** the FVD is **388.45**, which is very close to the full-pass AFB (375.12) and **still significantly outperforms both the original Wan2.1-I2V (449.68) and Wan2.1-FLF2V (413.68)**. In this setting, the inference time increases by only **35%** compared to the baseline (27 min vs. 20 min).
>
>    **Table 1: Comparison of average LPIPS scores (lower is better) between Forward ($I_0 \to I_{N-1}$) and Reverse ($I_{N-1} \to I_0$) generation across 436 videos.**
>
>    | Strategy | Metric | Stop Step ($K$) | Time (min) | Overhead | FVD $\downarrow$ | LPIPS $\downarrow$ | SSIM $\uparrow$ | PSNR $\uparrow$ | GPT-4o $\uparrow$ | Gemini $\uparrow$ |
>    | :--- | :--- | :---: | :---: | :---: | :---: | :---: | :---: | :---: | :---: | :---: |
>    | AFB | Q function(LPIPS) | 10 | 25 | +25% | 476.52 | 0.24 | 0.79 | 28.03 | 65.74 | 67.23 |
>    | AFB | Q function(LPIPS) | 25 | 31 | +55% | 428.96 | 0.20 | 0.88 | 31.74 | 81.53 | 82.64 |
>    | AFB | Q function(LPIPS) | 40 | 37 | +85% | 392.53 | 0.18 | 0.92 | 33.67 | 84.62 | 84.93 |
>
>    **Table 2: Performance of Fast-AFB strategy. At step 15, Fast-AFB achieves an optimal balance between performance and efficiency**
>
>    | Strategy | Metric | Stop Step ($K$) | Time (min) | Overhead | FVD $\downarrow$ | LPIPS $\downarrow$ | SSIM $\uparrow$ | PSNR $\uparrow$ | GPT-4o $\uparrow$ | Gemini $\uparrow$ |
>    | :--- | :--- | :---: | :---: | :---: | :---: | :---: | :---: | :---: | :---: | :---: |
>    | Fast-AFB | Q function(LPIPS) | 5 | 23 | +15% | 412.73 | 0.19 | 0.90 | 32.74 | 82.53 | 81.64 |
>    | Fast-AFB | Q function(LPIPS) | **15** | **27** | **+35%** | **388.45** | **0.18** | **0.93** | **33.68** | **85.32** | **86.13** |
>    | Fast-AFB | Q function(LPIPS) | 40 | 37 | +85% | 379.35 | 0.16 | 0.96 | 34.81 | 87.59 | 87.30 |
>
>    2.) **Conclusion**: By evaluating the predicted clean frame, we can robustly identify effective anchors even at early stages, offering users a flexible trade-off between quality and latency.
>
> 3. **Anchor Mirroring Formula:** As detailed in our Response to Weakness 1, the mirroring formula is not a tunable heuristic parameter but a logical deduction based on the universal "information attenuation, which is a systematic issue governed by model architecture and error accumulation, independent of generation direction. Our supplementary statistical experiments confirm that the quality breakdown points for forward and reverse generation align consistently (Frame 56 vs. Frame 55), validating the robustness of this mirroring assumption without the need for manual tuning. For the detailed theoretical explanation and quantitative verification, please refer to the "Response to Weakness 1" above.

---

> ### Author Response · Authors · 2025-11-24
> **Response to Reviewer hmuv**
>
> > **Response to Weakness 3 (Visual Quality Trade-offs in Extreme Cases):**
>
> Thank you for acknowledging our strong experimental results and for the insightful identification of the method's limitations.
>
> We fully agree with your assessment: while AFB achieves significant gains in temporal coherence, addressing physical implausibility in extreme scenarios indeed remains a hard limitation that has not been fully resolved.
>
> 1. Limitation: As illustrated in **Appendix D and Fig. 11**, AFB operates as a **training-free** plugin. While it effectively optimizes generation trajectories to enhance video coherence, its performance upper bound is inherently **constrained by the capabilities of the base I2V model**. Consequently, residual artifacts (e.g., limb distortions during non-rigid deformation) may persist in complex scenarios if the base model itself fails to capture the underlying physics correctly.
>
> 2. We accept the suggestion and will explicitly emphasize in the revised "Limitations" section that the dependency on the base model's capability constitutes a hard limitation of our method. To address this limitation, we plan to explore extending the AFB framework in future work by integrating motion-aware techniques or introducing additional physical constraint guidance, trying to proactively handle these challenging scenarios, thereby further improving physical plausibility and mitigating distortions.

---

> ### Author Response · Authors · 2025-11-24
> **Response to Reviewer hmuv**
>
> > **Response to Question 1 (Theoretical Support and Alternative Anchor Selection):**
>
> Thank you for the insightful question regarding the theoretical justification and robustness of our anchor selection strategy. To address this, we conduct supplementary experiments evaluating alternative strategies (e.g., fixed position insertion, different evaluation metrics) and analyze potential failure cases.
>
> 1. Comparison with Alternative Strategies:
>
>    1.) Fixed vs. Adaptive Anchor Position: We evaluated inserting anchor frames at fixed relative positions (e.g., $\alpha = 1/2, 1/4, 1/6$, ...). Empirical results demonstrate that our **adaptive strategy consistently outperforms all fixed strategies**.
>
>    | $\alpha$ positions | FVD $\downarrow$ | LPIPS $\downarrow$ | SSIM $\uparrow$ | PSNR $\uparrow$ |
>    | :---: | :---: | :---: | :---: | :---: |
>    | 1/2 | 447.30 | 0.25 | 0.84 | 32.57 |
>    | 1/3 | 424.36 | 0.22 | 0.87 | 33.19 |
>    | 1/4 | 386.12 | 0.17 | 0.93 | 34.51 |
>    | 1/5 | 418.11 | 0.21 | 0.86 | 33.01 |
>    | 1/6 | 415.67 | 0.20 | 0.88 | 33.93 |
>    | Adaptive Anchor Selection(Ours) | **375.12** | **0.16** | **0.97** | **35.41** |
>
>    2.) Alternative Criterion (CLIP vs. LPIPS): We compared the performance of **CLIP feature distance** and **LPIPS** as selection criteria. Our experiments reveal that **LPIPS consistently outperforms CLIP**. For instance, at $t=40$, LPIPS achieves an FVD of **392.53**, compared to 394.13 for CLIP.  We attribute this to the fundamental difference in metric objectives. CLIP prioritizes **high-level semantic alignment**, making it insensitive to artifacts like jitter or ghosting which disrupt visual continuity but often preserve semantic meaning. In contrast, LPIPS focuses on **perceptual similarity**, making it inherently more sensitive to the visual discontinuities. Therefore, LPIPS serves as a more precise proxy for "temporal coherence" in this specific task.
>
>
>
>  **Table 1: Performance comparison of Original AFB using different metrics and stop steps. Evaluating the noisy latent $x_t$ directly at early stages leads to instability for both metrics.**
>
>    | Strategy | Metric | Stop Step ($K$) | Time (min) | Overhead | FVD $\downarrow$ | LPIPS $\downarrow$ | SSIM $\uparrow$ | PSNR $\uparrow$ | GPT-4o $\uparrow$ | Gemini $\uparrow$ |
>    | :--- | :--- | :---: | :---: | :---: | :---: | :---: | :---: | :---: | :---: | :---: |
>    | AFB | Q function(LPIPS) | 10 | 25 | +25% | 476.52 | 0.24 | 0.79 | 28.03 | 65.74 | 67.23 |
>    | AFB | Q function(LPIPS) | 25 | 31 | +55% | 428.96 | 0.20 | 0.88 | 31.74 | 81.53 | 82.64 |
>    | AFB | Q function(LPIPS) | 40 | 37 | +85% | 392.53 | 0.18 | 0.92 | 33.67 | 84.62 | 84.93 |
>    | AFB | CLIP | 10 | 25 | +25% | 477.65 | 0.24 | 0.78 | 27.76 | 67.58 | 68.43 |
>    | AFB | CLIP | 25 | 31 | +55% | 429.83 | 0.21 | 0.87 | 30.54 | 83.58 | 83.41 |
>    | AFB | CLIP | 40 | 37 | +85% | 394.13 | 0.19 | 0.90 | 31.57 | 82.73 | 81.69 |
>
>   **Table 2: Performance of the optimized Fast-AFB strategy using predicted clean frames $\hat{x}_0$. At step 15, LPIPS achieves an optimal balance between performance and efficiency**
>
>    | Strategy | Metric | Stop Step ($K$) | Time (min) | Overhead | FVD $\downarrow$ | LPIPS $\downarrow$ | SSIM $\uparrow$ | PSNR $\uparrow$ | GPT-4o $\uparrow$ | Gemini $\uparrow$ |
>    | :--- | :--- | :---: | :---: | :---: | :---: | :---: | :---: | :---: | :---: | :---: |
>    | Fast-AFB | Q function(LPIPS) | 5 | 23 | +15% | 412.73 | 0.19 | 0.90 | 32.74 | 82.53 | 81.64 |
>    | Fast-AFB | Q function(LPIPS) | 15 | 27 | +35% | 388.45 | 0.18 | 0.93 | 33.68 | 85.32 | 86.13 |
>    | Fast-AFB | Q function(LPIPS) | 40 | 37 | +85% | 379.35 | 0.16 | 0.96 | 34.81 | 87.59 | 87.30 |
>    | Fast-AFB | CLIP | 5 | 23 | +15% | 415.68 | 0.19 | 0.89 | 32.31 | 81.67 | 82.04 |
>    | Fast-AFB | CLIP | 15 | 27 | +35% | 391.54 | 0.18 | 0.92 | 33.57 | 85.44 | 85.69 |
>    | Fast-AFB | CLIP | 40 | 37 | +85% | 383.20 | 0.17 | 0.94 | 34.56 | 86.32 | 87.20 |
>
> 2. Theoretical Support & Failure Cases:
>
>    1.) **Theoretical Insight:** As detailed in our **Response to Weakness 1**, the optimality of the "mirrored LPIPS breakpoint" is not coincidental but is grounded in the **"information attenuation"** inherent to the model architecture. The breakdown points in forward and reverse generation consistently align due to the systematic decay of boundary constraints.
>
>    2.) **Failure Cases:** As discussed in **Appendix D** and the **Limitation** section, our method operates as a plug-and-play module and is therefore bounded by the capabilities of the base model. In extreme scenarios (e.g., complex non-rigid motion or severe occlusion), if the base model fails to capture the underlying physics, the generated video may still exhibit residual physical implausibility.

---

> ### Author Response · Authors · 2025-11-24
> **Response to Reviewer hmuv**
>
> > **Response to Question 2 (Impact and Selection of Hyperparameters):**
>
> Thank you for your detailed and insightful questions.
>
> 1. As detailed in our **Response to Weakness 2**, we explored the impact of selecting different denoising phases. While the original method uses a stable phase (e.g., $t=45$), our new **Fast-AFB** strategy demonstrates that by evaluating the **predicted clean frame $\hat{x}\_0$**, we can robustly identify effective anchors even at early stages (e.g., **$t=15$**). This selection achieves an optimal balance, maintaining performance comparable to the full-pass method (FVD 388.45 vs. 375.12) while reducing inference overhead to just 35%.
>
> 2. **Window Width for Local Averaging:&#x20;**&#x57;e select the minimal window size (adjacent frames) based on the nature of the LPIPS metric. Since LPIPS is inherently robust to high-frequency noise and sensitive to visual and structural differences,  a minimal window is sufficient to capture abrupt "semantic breaks." Increasing the window size would unnecessarily smooth out these critical local variations, potentially obscuring the precise breakdown point. **Please refer to "Response to Weakness 2" for the detailed rationale.**
>
> 3. **Anchor Selection Heuristics:** The "mirroring" heuristic is not an arbitrary choice but a logical deduction based on the **universal mechanism of "information attenuation."** Our statistical analysis confirms that the breakdown points in forward and reverse generation consistently align (Frame 56 vs. Frame 55), validating the robustness of this heuristic without the need for manual tuning. **For the theoretical explanation and quantitative verification, please refer to "Response to Weakness 1" above.**

---

### Official Review · Reviewer_zLy5 · 2025-10-31

**Soundness:** 3
**Presentation:** 3
**Contribution:** 3
**Rating:** 6
**Confidence:** 4

**Summary:**

This paper addresses semantic degradation and temporal inconsistency in first-last frame video generation (FLF2V), where models must generate a video given only the start and end frames. The authors identify "information attenuation" as a key issue, where semantic guidance from the boundary frames weakens towards the middle of the sequence. To solve this, the paper introduces Anchor Frame Bridging (AFB), a novel, plug-and-play, and training-free method. AFB operates in two stages: The method first performs a full *reverse* generation pass (from last frame to first frame) to create a "candidate set" of frames. It identifies the frame with the maximum temporal incoherence (peak LPIPS) in this reversed sequence at a normalized position $\alpha$. It then selects an anchor frame from the *mirrored* position ($1-\alpha$) of this candidate set. The final video is synthesized by conditioning the base I2V model on the first frame, the last frame, and this new anchor frame at its designated position $\alpha$, using an indicator mask to guide the diffusion process. The authors created a new dataset of 436 video pairs and applied AFB to two base models (Wan2.1, Hunyuan Video). Results show quantitative improvements (e.g., 16.58% in FVD on Wan2.1-I2V) and improved qualitative coherence in a user study.

**Strengths:**

- The core idea of AFB is novel and clever. Using a reversed-generation pass to identify a point of failure ($\alpha$) and then applying a "mirror position" heuristic ($1-\alpha$) to select a high-quality anchor frame is an elegant, non-trivial solution to the problem of *what* to use as an anchor and *where* to place it.
- The method is training-free and plug-and-play. This is a significant practical advantage, allowing it to enhance existing I2V models for the FLF2V task without any costly fine-tuning.
- The paper is very well-written and easy to follow. Figure 1 and 2 provide an excellent, intuitive visualization of both the "information attenuation" problem and the proposed two-stage solution.
- The authors validate their method on two different base models and use a comprehensive set of metrics (FVD, LPIPS, PSNR/SSIM, MLLM scores, and a user study) to build a robust case for the method's effectiveness. The ablation studies (e.g., on anchor frame count in Fig. 5a) are valuable and clearly justify the final design.

**Weaknesses:**

1. The most significant weakness is the computational overhead. The "Adaptive Anchor Frame Selection" stage requires a full, additional reverse generation pass (e.g., 50 diffusion steps) just to find the anchor frame. As shown in Table 3, this effectively doubles the inference time (e.g., from 20 to 41 minutes for Wan2.1). This high cost severely limits the method's practical utility.
2. The ablation study (Fig. 5a and user study in Fig. 12) reveals that using a single anchor frame ($k=1$) is optimal, and $k>1$ degrades performance due to "conflicting guidance." This is a major limitation. It strongly suggests the method is brittle and cannot scale to longer, more complex videos with multiple distinct actions or scene changes, which would naturally require multiple anchor points to maintain coherence.
3. The entire anchor selection mechanism (finding max LPIPS in the reverse pass and mirroring the position $\alpha \rightarrow 1-\alpha$) is a clever heuristic, but it is not a learned or guaranteed-optimal strategy. It relies on an implicit assumption that the motion and degradation are somewhat symmetrical, which may not hold for many real-world scenarios (e.g., a video of a car accelerating and then coasting).
4. The paper positions AFB as a "plug-and-play" method. However, the experimental comparison in Table 1 is primarily against the base models themselves (Wan2.1-I2V, Hunyuan-I2V) or fully-trained methods (Wan2.1-FLF2V). The comparison lacks other relevant *training-free* or *guidance-based* video interpolation or editing methods that could also be considered "plug-and-play." This makes it difficult to assess AFB's performance relative to its true peers.
5. While the creation of a new dataset is commendable, the size (N=436) is very small for a video generation benchmark. This limited diversity makes it difficult to assess generalizability and raises concerns that the method's findings (especially the $k=1$ optimality) might be an artifact of the specific data distribution (e.g., short, single-action clips).

**Questions:**

1. The doubled inference time is a major drawback. Have the authors explored cheaper approximations for anchor selection? For instance, Figure 5b shows LPIPS is unstable in early timesteps. Could a different, more stable metric (e.g., CLIP feature distance) reliably estimate the failure point $\alpha$ at a much earlier, cheaper timestep (e.g., $t=10$ or $t=20$) instead of requiring a near-full reverse pass ($t=45$)?
2. The finding that $k>1$ anchor frames degrades performance is counter-intuitive and the most significant concern for scalability. Does this imply a fundamental limitation in the "anchor frame-guided generation" step, which cannot resolve conflicting semantic guidance from multiple anchors? How do you envision this method scaling to longer videos (e.g., 300 frames) where a single anchor is clearly insufficient?
3. $\alpha \rightarrow 1-\alpha$ "mirror" heuristic is clever. But what are its failure modes? Have you analyzed cases where the motion is asymmetric (e.g., "object appears" vs. "object disappears")? In such cases, the point of max LPIPS in the reverse pass might be semantically unrelated to the *ideal* anchor location for the forward pass.
4. The failure case analysis in Appendix D.1 is appreciated and shows failures inherited from the base models. Have you identified any failure cases *introduced* by AFB itself? That is, an example where the base model produces a coherent video, but the insertion of the selected anchor frame (which comes from a different generation pass) *causes* a new temporal break, artifact, or semantic mismatch?
5. Related to Weakness #5, is it possible that the $k=1$ optimality is an artifact of your small dataset? How would you expect this to hold on a more diverse, large-scale benchmark with more complex, multi-stage motions?

---

> ### Author Response · Authors · 2025-11-24
> **Response to Reviewer zLy5**
>
> **We sincerely thank reviewer for the constructive feedback and the positive comments.** We are delighted that the reviewer finds our core idea to be **“novel and clever”** and characterizes our **anchor selection strategy** as an **“elegant, non-trivial solution”**. We also appreciate the recognition of our method’s **“significant practical advantage”** as a training-free, plug-and-play module, and the acknowledgement that we built a **“robust case”** for effectiveness through comprehensive metrics.  In the following, we provide detailed responses to the weaknesses and questions raised. We hope our response fully resolves your concerns.
>
> > **Response to Weakness 1 (Computational Overhead):**
>
> **We sincerely thank you for highlighting the computational overhead and its impact on practical utility.** We acknowledge that the full reverse pass doubles the inference time. To address this and enhance practicality, we conducted comprehensive evaluations and propose the optimized **Fast-AFB** strategy.
>
> 1. **Methodology (Fast-AFB):** Specifically, we designed an acceleration scheme based on the suggested truncated denoising. However, directly evaluating the noisy latent $x_t$ in early reverse stages is unreliable for anchor selection. To address this, we refined the metric to evaluate the **predicted clean frame** $\hat{x}_0$ derived from the current step $x_t$, rather than $x_t$ itself. The prediction formula is:
> $$\hat{x} _ 0 = \frac{x _ t - \sqrt{1-\bar{\alpha} _ t}\epsilon _ \theta(x _ t, t)}{\sqrt{\bar{\alpha} _ t}}$$
>
> 2. **Experimental Results:** As shown in the table below, we evaluate the trade-offs at various stop steps. **Notably, terminating at timestep 15 (**$t=15$**) achieves an optimal balance:** the FVD is 388.45, which is very close to the full-pass AFB (375.12) and still significantly outperforms both the original Wan2.1-I2V (449.68) and Wan2.1-FLF2V (413.68), while the inference time increases by only **35%** compared to the baseline (27 min vs. 20 min).
>
> **The Fast-AFB strategy offers users a flexible "Efficiency-Quality Trade-off":** users can prioritize maximum fidelity with the full AFB, or opt for Fast-AFB to achieve substantial performance gains over baselines with only a marginal increase in time cost.
>
> **Table 1: Comparison of average LPIPS scores (lower is better) between Forward ($I_0 \to I_{N-1}$) and Reverse ($I_{N-1} \to I_0$) generation across 436 videos.**
>
> | Strategy | Metric | Stop Step ($K$) | Time (min) | Overhead | FVD $\downarrow$ | LPIPS $\downarrow$ | SSIM $\uparrow$ | PSNR $\uparrow$ | GPT-4o $\uparrow$ | Gemini $\uparrow$ |
> | :--- | :--- | :---: | :---: | :---: | :---: | :---: | :---: | :---: | :---: | :---: |
> | AFB | Q function(LPIPS) | 10 | 25 | +25% | 476.52 | 0.24 | 0.79 | 28.03 | 65.74 | 67.23 |
> | AFB | Q function(LPIPS) | 25 | 31 | +55% | 428.96 | 0.20 | 0.88 | 31.74 | 81.53 | 82.64 |
> | AFB | Q function(LPIPS) | 40 | 37 | +85% | 392.53 | 0.18 | 0.92 | 33.67 | 84.62 | 84.93 |
> | AFB | Q function(LPIPS) | 45 | 41 | +105% | 375.12 | 0.16 | 0.97 | 35.41 | 88.64 | 89.35 |
>
> **Table 2: Performance of Fast-AFB strategy. At step 15, Fast-AFB achieves an optimal balance between performance and efficiency**
>
> | Strategy | Metric | Stop Step ($K$) | Time (min) | Overhead | FVD $\downarrow$ | LPIPS $\downarrow$ | SSIM $\uparrow$ | PSNR $\uparrow$ | GPT-4o $\uparrow$ | Gemini $\uparrow$ |
> | :--- | :--- | :---: | :---: | :---: | :---: | :---: | :---: | :---: | :---: | :---: |
> | Fast-AFB | Q function(LPIPS) | 5 | 23 | +15% | 412.73 | 0.19 | 0.90 | 32.74 | 82.53 | 81.64 |
> | Fast-AFB | Q function(LPIPS) | **15** | **27** | **+35%** | **388.45** | **0.18** | **0.93** | **33.68** | **85.32** | **86.13** |
> | Fast-AFB | Q function(LPIPS) | 40 | 37 | +85% | 379.35 | 0.16 | 0.96 | 34.81 | 87.59 | 87.30 |

---

> ### Author Response · Authors · 2025-11-24
> **Response to Reviewer zLy5**
>
> > **Response to Weakness 2 (Scalability and Multi-Anchor Performance):**
>
> Thank you for the critical question regarding the performance drop at $K > 1$ and the method's scalability.
>
> 1. **First, we wish to clarify the concern regarding the method being "brittle."** In our experiments, while $K=1$ achieved optimal performance, AFB demonstrates strong robustness even in sub-optimal configurations. As shown in Figure 5 (a) in main text, the original baseline Wan2.1-I2V has an FVD of 449.68. In contrast, the FVD is **386.94 for $K=2$** and **397.50 for $K=3$**. The performance drop at $K>1$ is relative to the optimal $K=1$ configuration. At $K=2$, the method still outperforms the original baseline; at $K=3$, the model effectively addresses video coherence issues (with FVD and LPIPS superior to the baseline), though it introduces slight pixel-level distortion (lower SSIM and PSNR). This demonstrates that the AFB mechanism is inherently robust and effective, successfully fulfilling its primary design objective of resolving semantic continuity breaks.
>
> 2. Second, the performance degradation at $K > 1$ stems from a mismatch between "constraint density" and "video duration," rather than a fundamental limitation of our method. In our current short-video experiments (< 5s), inserting multiple anchors results in **excessive temporal constraint density**. This over-constrains the base model's motion manifold, triggering conflicting guidance. **The conflict arises because the temporal transition windows between anchors are insufficient, not because our method is incapable of handling multiple constraints.**
>
> 3. **To further validate this, we conduct supplementary experiments ( see Appendix J Figure 21):** we utilize **Jimeng AI** to generate **10-second long videos**.  Results show that in long-video generation, **a multi-anchor strategy (inserting 2 or 3 frames) outperforms the single-anchor approach.** As the number of anchors increases, the coherence of intermediate processes in long-range videos improves. In the future, addressing longer videos and more challenging complex scenarios, we will extend AFB to explore adaptive multi-anchor insertion methods.

---

> ### Author Response · Authors · 2025-11-24
> **Response to Reviewer zLy5**
>
> > **Response to Weakness 3 ("Mirror" Heuristic and Asymmetric Motion):**
>
> Thank you for your keen insight regarding the anchor selection mechanism. We acknowledge that real-world dynamics (e.g., a car accelerating and then coasting) are indeed physically asymmetric.
>
> However, we wish to clarify that **the "symmetry" of "quality degradation" in generative models is not equivalent to the physical symmetry of object motion speed.** The "positional consistency" of quality degradation in both forward and reverse passes is governed by "information attenuation," not by the physical object motion.
>
> 1. **Theoretical Explanation:** "Information attenuation" is a systematic issue driven by model architecture and error accumulation. Regardless of the generation direction (forward or reverse) or the speed of the object, the semantic constraints from first-last frames naturally decay over time.
>
>    1.) **Mechanism:** As the distance from the conditioning frame increases, this weakening constraint allows minor visual flaws to go uncorrected. These tiny deviations propagate, accumulate, and amplify frame-by-frame, eventually causing the generated content to deviate from a plausible trajectory and triggering a continuity break at a specific "semantic vacuum" .
>
>    2.) **Directional Independence:** Although reverse generation changes the direction, the mechanisms of "constraint decay" and "error accumulation" persist, inevitably leading to a continuity break at a **similar distance from the input frame**.
>
> 2. **Quantitative Experiment:** To verify this assumption, we conduct a quantitative experiment across our dataset, encompassing diverse motion types. We calculated the frame-wise average LPIPS scores for both forward generation ($I_0 \to I_{N-1}$) and reverse generation ($I_{N-1} \to I_0$) . The statistical results (**summarized in the table below and detailed in Appendix G Figure 18**) reveal that the quality breakdown points (peak LPIPS) for forward and reverse generation align remarkably well (Frame 56 vs. Frame 55), with a deviation of only 1 frame. This provides strong statistical evidence that the "positional consistency" between forward and reverse generation is robust.
>
> **Table: Comparison of average LPIPS scores (lower is better) between Forward ($I _ 0 \to I _ {N-1}$) and Reverse ($I _ {N-1} \to I _ 0$) generation across 436 videos.**
>
> | Frame ID | 0 | 5 | 10 | 15 | 20 | 25 | 30 | 35 | 40 | 45 | 50 | **55** | 60 | 65 | 70 | 75 | 80 |
> | :--- | :---: | :---: | :---: | :---: | :---: | :---: | :---: | :---: | :---: | :---: | :---: | :---: | :---: | :---: | :---: | :---: | :---: |
> | Forward | 0.03 | 0.055 | 0.06 | 0.07 | 0.07 | 0.076 | 0.08 | 0.085 | 0.09 | 0.103 | 0.11 | **0.163** | 0.14 | 0.126 | 0.088 | 0.079 | 0.067 |
> | Reverse | 0.04 | 0.06 | 0.059 | 0.063 | 0.069 | 0.074 | 0.079 | 0.084 | 0.087 | 0.127 | 0.179 | **0.22** | 0.152 | 0.152 | 0.135 | 0.114 | 0.086 |
>
> 3. **Qualitative Verification: As illustrated in Appendix K Figure 22** (case: car accelerating and then coasting), the decay of semantic constraints manifests similarly in both forward and reverse generations, regardless of whether the car is fast or slow.  Specifically, both directions exhibit motion discontinuities, such as the car failing to initiate acceleration naturally. In contrast, after applying AFB, the anchor frame effectively bridges this semantic gap, rendering the acceleration process significantly smoother and physically coherent.
>
> 4. **Conclusion:** The evidence above confirms that the breakdown points in forward and reverse generation consistently align, ensuring the effectiveness of our heuristic across diverse motion types.

---

> ### Author Response · Authors · 2025-11-24
> **Response to Reviewer zLy5**
>
> > **Response to Weakness 4 (Comparisons with other "Plug-and-Play" method):**
>
> Thank you for the valuable suggestion regarding baseline comparisons.
>
> 1. **Comparsion with ViBiDSampler :** We respectfully point out that **ViBiDSampler** (Yang et al., 2025a), which is already included in Table 1, serves as a representative **training-free, guidance-based method** relying on bidirectional diffusion sampling. As shown in Table 1, AFB significantly outperforms ViBiDSampler in both FVD (**375.12** vs. 426.15) and PSNR (**35.41** vs. 33.08). This strongly evidences that our proposed anchor-frame guidance is more effective for long-term consistency.
>
> 2. **Additional Comparisons (FCVG & MoG):** We have added detailed comparisons with FCVG \[1] and MoG \[2] in Appendix I. While relevant, AFB demonstrates distinct advantages as a plug-and-play solution:
>
>    1. **vs. FCVG \[1]:** FCVG relies on linear interpolation of explicit conditions (e.g., matched lines/pose) and requires **fine-tuning extra modules**. Its linear assumption often leads to rigid motion in complex scenarios. In contrast, AFB is **training-free**; it utilizes reverse generation to adaptively detect "semantic breakpoints," preserving the base model's generative freedom for natural, non-linear motion.
>
>    2. **vs. MoG \[2]:** MoG relies on intermediate optical flow and requires **fine-tuning spatial layers** to correct warping errors. Its performance is bounded by flow estimation accuracy. In contrast, AFB avoids pixel-level flow dependency and directly targets semantic degradation.
>
>    3. **Quantitative Results:** As shown in the table below, **AFB significantly outperforms both FCVG and MoG**, achieving superior temporal coherence (lowest FVD) and visual fidelity (highest PSNR). Notably, while FCVG and MoG typically generate short clips (approx. 1-2s), AFB maintains better consistency even when generating longer, complex videos (approx. 5s).
>
>    **Table: Quantitative comparison with related works.** AFB significantly outperforms both FCVG and MoG across all metrics.
>
>    | Baseline Models | LPIPS $\downarrow$ | FVD $\downarrow$ | SSIM $\uparrow$ | PSNR $\uparrow$ |
>    | :--- | :---: | :---: | :---: | :---: |
>    | FCVG | 0.19 | 438.83 | 0.86 | 33.62 |
>    | MoG  | 0.26 | 421.53 | 0.69 | 18.41 |
>    | ViBiDSampler | 0.19 | 426.15 | 0.90 | 33.08 |
>    | **Wan2.1 + AFB (Ours)** | **0.16** | **375.12** | **0.97** | **35.41** |
>
> * \[1] Zhu T, Ren D, Wang Q, et al. Generative inbetweening through frame-wise conditions-driven video generation\[C]//Proceedings of the Computer Vision and Pattern Recognition Conference. 2025: 27968-27978.
>
> * \[2] Zhang G, Zhu Y, Cui Y, et al. Motion-aware generative frame interpolation\[J]. arXiv preprint arXiv:2501.03699, 2025.

---

> ### Author Response · Authors · 2025-11-24
> **Response to Reviewer zLy5**
>
> > **Response to Weakness 5 (Dataset Diversity and Generalizability):**
>
> **Thank you for the valuable feedback regarding dataset construction.** Regarding your concern that the dataset might be limited to "short, single-action clips," we wish to clarify that our benchmark is designed to cover diverse scenes and **heterogeneous motion patterns** to ensure comprehensive evaluation.
>
> 1. **Diverse Categories:** As illustrated in Fig. 3, our dataset is not confined to a single distribution but spans a wide range of categories, including Scene/Natural Landscape (26%), Movie/Narrative (21%), and Sports (20%).
>
> 2. **Heterogeneous Motion Patterns:** The dataset incorporates complex non-rigid deformations (e.g., humans and animals from **DAVIS**), general object dynamics (e.g., vehicles from online videos), and significant camera movements (e.g., pans and zooms from **RealEstate10K** ).
>
> 3. **Challenging Dynamics:** Crucially, during dataset construction, we explicitly filtered for samples exhibiting **"significant visual variation"** between the first and last frames. This ensures the inclusion of challenging scenarios involving large-scale dynamics, rather than simple, static transitions. Therefore, the findings derived from this benchmark reflect the method's performance across a broad spectrum of realistic video generation tasks.
>
> 4. **Verification via Robustness Analysis (Physical & Complexity)**: To further validate the method's robustness in diverse scenarios, we conduct supplementary experiments on targeted challenging scenarios（**see Appendix F Figure 17 and Appendix H Figure 19**)
>
>    1.) **Irreversible Physical Events:** In scenarios violating reverse physics (e.g., **glass shattering, smoke dissipating**), our experiments confirm that AFB successfully extracts valid anchors from the reliable initial segments of the reverse pass, guiding coherent generation even when reverse global physical causality is challenging.
>
>    2.) **High Complexity Variance:** In experiments with high dynamic variance, such as **complex motion (e.g., Parkour)** and **scene transitions (e.g., corridor to outdoors)**, AFB maintains robust performance, effectively bridging the complexity gap between significantly different boundary frames.
>
> 5. **Conclusion:** Collectively, these extensive evaluations across diverse categories, physical laws, and complexity levels confirm that AFB's effectiveness is a **generalizable capability**, not an artifact of specific data distributions.

---

> ### Author Response · Authors · 2025-11-24
> **Response to Reviewer zLy5**
>
> > **Response to Question 1 (Computational Overhead):**
>
> Thank you for the insightful suggestion regarding cheaper approximations and alternative metrics like CLIP feature distance.
>
> 1. **Exploration of Cheaper Approximations (Fast-AFB):  We have successfully explored and implemented a cheaper approximation.** As detailed in our response to **Weakness 1**, we **propose the Fast-AFB strategy**. Instead of a near-full reverse pass, we utilize **truncated denoising** at early timesteps (e.g., $t=15$). Crucially, to solve the instability issue shown in Fig. 5b, we shift the evaluation target from the noisy latent $x_t$ to the **predicted clean frame** $\hat{x}_0$. This optimization achieves robust anchor selection at $t=15$ with only a **35\% increase** in inference time (vs. 105\% originally), effectively addressing the cost concern. **Please refer to the "Response to Weakness 1" above for the detailed methodology and results.**
>
> 2. Comparison between different metrics: **Following your suggestion, we have compared the performance of CLIP feature distance and LPIPS for anchor selection (see Table below).** Our findings are as follows:
>
>    1. **Impact of Noise (**$x _ t$ vs $\hat{x} _ 0$**):** The instability at early timesteps stems from the heavy noise in $x_t$, which disrupts *both* metrics. However, when evaluating the predicted clean frame $\hat{x}_0$, both metrics stabilize significantly.
>
>    2. **Superiority of LPIPS:** LPIPS consistently outperforms CLIP, regardless of whether the evaluation is performed on the noisy latent $x_t$ or the predicted clean frame $\hat{x}_0$ (e.g., on $\hat{x}_0$ at $t=15$, FVD is 388.45 for LPIPS vs. 391.54 for CLIP). We attribute this to the fundamental difference in metric objectives: CLIP prioritizes **high-level semantic alignment**, making it insensitive to artifacts like jitter or ghosting which disrupt visual continuity but often preserve semantic meaning. In contrast, LPIPS focuses **perceptual similarity**, making it inherently more sensitive to visual discontinuities. Therefore, LPIPS serves as a more precise proxy for "temporal coherence" in this specific task.
>
> **Table 1: Performance comparison of Original AFB using different metrics and stop steps. Evaluating the noisy latent $x_t$ directly at early stages leads to instability for both metrics.**
>
>    | Strategy | Metric | Stop Step ($K$) | Time (min) | Overhead | FVD $\downarrow$ | LPIPS $\downarrow$ | SSIM $\uparrow$ | PSNR $\uparrow$ | GPT-4o $\uparrow$ | Gemini $\uparrow$ |
>    | :--- | :--- | :---: | :---: | :---: | :---: | :---: | :---: | :---: | :---: | :---: |
>    | AFB | Q function(LPIPS) | 10 | 25 | +25% | 476.52 | 0.24 | 0.79 | 28.03 | 65.74 | 67.23 |
>    | AFB | Q function(LPIPS) | 25 | 31 | +55% | 428.96 | 0.20 | 0.88 | 31.74 | 81.53 | 82.64 |
>    | AFB | Q function(LPIPS) | 40 | 37 | +85% | 392.53 | 0.18 | 0.92 | 33.67 | 84.62 | 84.93 |
>    | AFB | CLIP | 10 | 25 | +25% | 477.65 | 0.24 | 0.78 | 27.76 | 67.58 | 68.43 |
>    | AFB | CLIP | 25 | 31 | +55% | 429.83 | 0.21 | 0.87 | 30.54 | 83.58 | 83.41 |
>    | AFB | CLIP | 40 | 37 | +85% | 394.13 | 0.19 | 0.90 | 31.57 | 82.73 | 81.69 |
>
>   **Table 2: Performance of the optimized Fast-AFB strategy using predicted clean frames $\hat{x}_0$. At step 15, LPIPS achieves an optimal balance between performance and efficiency**
>    | Strategy | Metric | Stop Step ($K$) | Time (min) | Overhead | FVD $\downarrow$ | LPIPS $\downarrow$ | SSIM $\uparrow$ | PSNR $\uparrow$ | GPT-4o $\uparrow$ | Gemini $\uparrow$ |
>    | :--- | :--- | :---: | :---: | :---: | :---: | :---: | :---: | :---: | :---: | :---: |
>    | Fast-AFB | Q function(LPIPS) | 5 | 23 | +15% | 412.73 | 0.19 | 0.90 | 32.74 | 82.53 | 81.64 |
>    | Fast-AFB | Q function(LPIPS) | 15 | 27 | +35% | 388.45 | 0.18 | 0.93 | 33.68 | 85.32 | 86.13 |
>    | Fast-AFB | Q function(LPIPS) | 40 | 37 | +85% | 379.35 | 0.16 | 0.96 | 34.81 | 87.59 | 87.30 |
>    | Fast-AFB | CLIP | 5 | 23 | +15% | 415.68 | 0.19 | 0.89 | 32.31 | 81.67 | 82.04 |
>    | Fast-AFB | CLIP | 15 | 27 | +35% | 391.54 | 0.18 | 0.92 | 33.57 | 85.44 | 85.69 |
>    | Fast-AFB | CLIP | 40 | 37 | +85% | 383.20 | 0.17 | 0.94 | 34.56 | 86.32 | 87.20 |

---

> ### Author Response · Authors · 2025-11-24
> **Response to Reviewer zLy5**
>
> > **Response to Question 2 (Scalability and Multi-Anchor Performance):**
>
> Thank you for raising the concern regarding the counter-intuitive performance drop at $K > 1$ and the method's scalability.
>
> 1. Multi-Anchor Performance: As detailed in our response to **Weakness 2**, the performance drop in short videos stems from **"excessive temporal constraint density"** rather than a fundamental limitation of the "anchor frame-guided generation" step. In short sequences (< 5s), inserting multiple anchors over-constrains the motion manifold, leaving insufficient temporal windows for natural transitions.
>
> 2. **Scalability to Longer Videos:** Regarding your question on scaling to longer videos, our supplementary experiments on **10-second videos** confirm that **a multi-anchor strategy (**$K=2, 3$**) outperforms the single-anchor approach** when the duration increases. This indicates that introducing multiple anchors is an effective strategy for maintaining coherence in longer sequences. In the future, to address even longer videos and more complex scenarios, we will extend AFB to explore adaptive multi-anchor insertion methods. **Please refer to the "Response to Weakness 2" above for the detailed experimental results and analysis.**
>
>
>
> > **Response to Question 3 ("Mirror" Heuristic and Asymmetric Motion):**
>
> Thank you for the insightful question regarding potential failure modes in asymmetric scenarios.
>
> 1. **Clarification on Asymmetry:** As detailed in our response to **Weakness 3**, we clarify that the "positional consistency" of quality degradation is governed by **"information attenuation"** (a systematic model characteristic driven by error accumulation), rather than the physical symmetry of the motion. Therefore, even in physically asymmetric cases (e.g., "object appears" or "acceleration vs. coasting"), the model loses track of boundary constraints at a **similar temporal position** in both directions.
>
> 2. **Robustness Verification:** Our quantitative experiment confirms that the breakdown points in forward and reverse generation **consistently align** (Frame 56 vs. Frame 55 on average), regardless of motion type. This demonstrates that the "mirror" heuristic relies on the consistent boundary of the model's generative capability, ensuring robustness. **Please refer to the "Response to Weakness 3" above for the detailed theoretical explanation and experimental verification.**

---

> ### Author Response · Authors · 2025-11-24
> **Response to Reviewer zLy5**
>
> > **Response to Question 4 (Failure Cases Introduced by AFB):**
>
> We thank the reviewer for this insightful question. We acknowledge the theoretical risk that **exceptionally poor reverse generation quality** (e.g., exhibiting severe distortion or semantic drift) could **yield a low-quality anchor.** In such cases, forcibly inserting this flawed anchor into the forward process might indeed disrupt the continuity that the baseline would otherwise have preserved.
>
> However, in our extensive experiments, such cases are **extremely rare**. This is primarily due to two reasons:
>
> 1. **Comparable Constraints:**  Although the generation direction differs, both forward and reverse processes operate under **equally constraints**: explicit visual conditioning from first-last frames and semantic guidance from text. Given the robust capabilities of modern base models, it is highly improbable that the reverse pass would suffer catastrophic degradation while the forward pass remains coherent, as both are grounded in the same underlying visual content.
>
> 2. **Correlation with Task Difficulty:** We observed that when the reverse pass fails to produce a valid anchor, it typically indicates that the motion is inherently too complex for the base model to handle. In these **challenging cases,** the baseline (pure forward generation) also struggles to produce high-quality coherent video. Therefore, even if AFB introduces a low-quality anchor in such instances, the primary bottleneck remains the **inherent limitations of the base model** in handling such extreme dynamics, rather than a flaw introduced by the AFB mechanism itself.
>
> In the rare cases where AFB selects a suboptimal anchor while the baseline performs adequately, this typically stems from the inherent stochasticity of the diffusion sampling process. We found that simply **updating the random seed and performing re-inference** is usually sufficient to retrieve a high-quality anchor and restore performance.
>
>
>
> > **Response to Question 5 ($K=1$ Optimality and Generalizability):**
>
> Thank you for the question regarding the optimality of $K=1$ and the method's performance on broader benchmarks.
>
> 1. Firstly, We wish to clarify that the optimality of $K=1$ is **not an artifact of limited dataset diversity or specific data distributions.** As detailed in our Response to Weakness 5, our dataset covers heterogeneous motion patterns and challenging dynamics. The consistent performance of $K=1$ across these varied cases confirms its robustness. Please refer to "Response to Weakness 5" above for more details.
>
> 2. The observed optimality of $K=1$ is fundamentally driven by the **video duration** rather than the dataset size. As discussed in **Response to Weakness 2**, inserting multiple anchors in short sequences creates excessive constraints, leading to conflicts. **Please refer to "Response to Weakness 2" above for the detailed mechanism.**
>
> 3. **We agree that for longer, more complex video generation tasks, introducing a multi-anchor strategy is necessary.** Our supplementary experiments on **10-second videos** (see Response to Weakness 2 and Appendix J Figure 20) confirm that multi-anchor strategies ($K>1$) significantly outperform single-anchor ones in long-range generations.
>
> 4. Guided by these findings, we will extend AFB to explore adaptive multi-anchor insertion methods, aiming to provide a general solution for long video generation in the future.

---

### Official Review · Reviewer_36Wi · 2025-11-01

**Soundness:** 2
**Presentation:** 3
**Contribution:** 1
**Rating:** 4
**Confidence:** 5

**Summary:**

The paper proposes Anchor Frame Bridging (AFB), a training-free, plug-and-play method to address semantic degradation and temporal inconsistency in first-last frame video generation (FLF2V). AFB consists of two core modules: (1) adaptive anchor frame selection, which uses reverse generation (swapped first/last frames + Qwen-generated prompts) and LPIPS-based quality scoring to identify high-semantic-consistency anchors; (2) anchor-guided generation, which injects anchors into the video sequence to propagate boundary semantics. The authors construct a 436-pair FLF2V dataset, validate AFB on Wan2.1 and Hunyuan Video models, and demonstrate improvements (16.58% FVD, 10.21% PSNR on Wan2.1-I2V) via quantitative metrics, MLLM evaluations (GPT-4o, Gemini), user studies, and qualitative comparisons.

**Strengths:**

1. Practical, training-free design: AFB avoids the high computational cost of retraining large video models, acting as a plug-and-play enhancement—critical for real-world FLF2V applications (e.g., video editing) where retraining is infeasible.

2. Rigorous validation: Results are supported by multi-faceted evaluation: standard metrics (FVD, PSNR, LPIPS), MLLM-based semantic assessment, user studies (52 participants), and ablations (anchor count, diffusion timesteps, prompt quality)—ensuring robustness.

3. Targeted problem diagnosis: The paper clearly links temporal inconsistency to sparse inter-frame attention (Fig.1a) and semantic drift, with visualizations (Fig.6,7) that directly demonstrate AFB’s ability to bridge boundary-to-intermediate semantics.

**Weaknesses:**

1. Significant computational overhead: AFB doubles inference time (e.g., Wan2.1: 20→41 min; Table 3) due to the reverse pass for anchor candidates. The paper provides no roadmap for optimization (e.g., truncated denoising, lightweight anchor generation) to mitigate this..

2. Limited edge-case improvement: AFB still struggles with extreme scenarios (non-rigid motion, severe occlusions; Fig.11) and relies on better base I2V models to resolve these—there is no exploration of extending AFB (e.g., dynamic anchor counts, motion-aware selection) to proactively address these failures.

3. Missing references and comparisons for first-last frame consistency works: The paper’s related work (Sec. 2) and experiments (Sec. 4) fail to reference or compare with [1] and [2]—works that presumably address the same first-last frame consistency problem. Without detailing how AFB’s technical approach (e.g., reverse generation, anchor selection) differs from [1]/[2] or how its performance (e.g., FVD, visual coherence) stacks against them, the paper’s positioning as a competitive solution for FLF2V is weakened.

4. Limited visual quality in supplementary materials: The supplementary material presents generated videos with limited quality, and the paper provides no direct comparison showing AFB outperforms [1] or [2] in visual fidelity. This lack of side-by-side visual evidence—critical for assessing FLF2V performance—makes it hard to verify AFB’s claimed advantages over these existing methods.

[1] Zhu T, Ren D, Wang Q, et al. Generative inbetweening through frame-wise conditions-driven video generation[C]//Proceedings of the Computer Vision and Pattern Recognition Conference. 2025: 27968-27978.

[2] Zhang G, Zhu Y, Cui Y, et al. Motion-aware generative frame interpolation[J]. arXiv preprint arXiv:2501.03699, 2025.

**Questions:**

Does the authors have plans to optimize the reverse pass—such as reducing denoising steps for anchor candidate generation or reusing latent features from the forward process—to lower latency, while ensuring the selected anchor frames still maintain high semantic consistency with the first/last frames?

---

> ### Author Response · Authors · 2025-11-24
> **Response to Reviewer 36Wi**
>
> We sincerely thank Reviewer for the insightful comments.  We are deeply appreciative that the reviewer recognizes our work as a **“practical, training-free design”** and a **“plug-and-play enhancement”** that is **“critical for real-world FLF2V applications”**. Furthermore, we are encouraged that the reviewer acknowledges our **“rigorous validation”** via multi-faceted evaluation and commends our **“targeted problem diagnosis”** regarding sparse inter-frame attention. In the following, we provide detailed responses to the weaknesses and questions raised. We hope our response fully resolves your concerns.
>
>
> > **Response to Weakness1 (Computational Overhead):**
>
> We sincerely thank the reviewer for highlighting the computational overhead and providing the constructive optimization roadmap. Inspired by these suggestions, we conducted comprehensive evaluations and propose the optimized **Fast-AFB** strategy.
>
> 1. **Methodology (Fast-AFB):** Specifically, we designed an acceleration scheme based on the suggested truncated denoising. However, directly evaluating the noisy latent $x_t$ in early reverse stages is unreliable for anchor selection. To address this, we refined the metric to evaluate the **predicted clean frame** $\hat{x}_0$ derived from the current step $x_t$, rather than $x_t$ itself. The prediction formula is:
> $$\hat{x} _ 0 = \frac{x _ t - \sqrt{1-\bar{\alpha} _ t}\epsilon _ \theta(x _ t, t)}{\sqrt{\bar{\alpha} _ t}}$$
>
> 2. **Experimental Results:** As shown in the table below, we evaluate the trade-offs at various stop steps. **Notably, terminating at timestep 15 (**$t=15$**) achieves an optimal balance:** the FVD is 388.45, which is very close to the full-pass AFB (375.12) and still significantly outperforms both the original Wan2.1-I2V (449.68) and Wan2.1-FLF2V (413.68), while the inference time increases by only **35%** compared to the baseline (27 min vs. 20 min).
>
> **The Fast-AFB strategy offers users a flexible "Efficiency-Quality Trade-off":** users can prioritize maximum fidelity with the full AFB, or opt for Fast-AFB to achieve substantial performance gains over baselines with only a marginal increase in time cost.
>
> **Table 1: Comparison of average LPIPS scores (lower is better) between Forward ($I_0 \to I_{N-1}$) and Reverse ($I_{N-1} \to I_0$) generation across 436 videos.**
>
> | Strategy | Metric | Stop Step ($K$) | Time (min) | Overhead | FVD $\downarrow$ | LPIPS $\downarrow$ | SSIM $\uparrow$ | PSNR $\uparrow$ | GPT-4o $\uparrow$ | Gemini $\uparrow$ |
> | :--- | :--- | :---: | :---: | :---: | :---: | :---: | :---: | :---: | :---: | :---: |
> | AFB | Q function(LPIPS) | 10 | 25 | +25% | 476.52 | 0.24 | 0.79 | 28.03 | 65.74 | 67.23 |
> | AFB | Q function(LPIPS) | 25 | 31 | +55% | 428.96 | 0.20 | 0.88 | 31.74 | 81.53 | 82.64 |
> | AFB | Q function(LPIPS) | 40 | 37 | +85% | 392.53 | 0.18 | 0.92 | 33.67 | 84.62 | 84.93 |
> | AFB | Q function(LPIPS) | 45 | 41 | +105% | 375.12 | 0.16 | 0.97 | 35.41 | 88.64 | 89.35 |
>
> **Table 2: Performance of Fast-AFB strategy. At step 15, Fast-AFB achieves an optimal balance between performance and efficiency**
>
> | Strategy | Metric | Stop Step ($K$) | Time (min) | Overhead | FVD $\downarrow$ | LPIPS $\downarrow$ | SSIM $\uparrow$ | PSNR $\uparrow$ | GPT-4o $\uparrow$ | Gemini $\uparrow$ |
> | :--- | :--- | :---: | :---: | :---: | :---: | :---: | :---: | :---: | :---: | :---: |
> | Fast-AFB | Q function(LPIPS) | 5 | 23 | +15% | 412.73 | 0.19 | 0.90 | 32.74 | 82.53 | 81.64 |
> | Fast-AFB | Q function(LPIPS) | **15** | **27** | **+35%** | **388.45** | **0.18** | **0.93** | **33.68** | **85.32** | **86.13** |
> | Fast-AFB | Q function(LPIPS) | 40 | 37 | +85% | 379.35 | 0.16 | 0.96 | 34.81 | 87.59 | 87.30 |

---

> ### Author Response · Authors · 2025-11-24
> **Response to Reviewer 36Wi**
>
> > **Response to Weakness 2 (Edge-case Improvement):**
>
> We appreciate the reviewer's insight regarding the limitations in edge cases.  We agree that in extreme scenarios (e.g., non-rigid motion, severe occlusions), AFB's performance is indeed bounded by the capabilities of the base model. **As a training-free, plug-and-play module, AFB is designed to address the common problem of video coherence degradation caused by information attenuation in FLF2V models.** Although limitations remain in extreme edge cases, quantitative results confirm that AFB successfully mitigates prevalent scene distortions and significantly boosts the semantic consistency and overall quality of the generated videos.
>
> We thank you for the valuable suggestions regarding extending AFB, and we will explore optimized schemes to better address these challenging tasks in the future:
>
> 1. **Dynamic anchor counts:** Our existing experiments indicate that for the short video generation currently performed, increasing the number of anchors ($K>1$) introduces over-constraints and may even lead to conflicting guidance. This proves that $K=1$ is the optimal balance between robustness and constraint density for current scenarios. However, we agree that for longer and more complex video generation, a multi-anchor strategy is necessary. In supplementary experiment as shown in **Appendix J Figure 21**, we utilized **Jimeng AI** to generate **10-second long videos**. Results show that  **a multi-anchor strategy (inserting 2 or 3 frames) outperforms the single-anchor approach**. As the number of anchors increases, the coherence of intermediate processes in long-range videos improves. In the future, we will explore a **dynamic anchor quantity mechanism** to introduce multiple anchors while maintaining reasonable constraint density, further satisfying the coherence requirements of long-range complex trajectories.
>
> 2. **Motion-aware selection:** Furthermore, we also plan to explore more **motion-sensitive anchor evaluation and selection mechanisms** (e.g., introducing motion-aware selection techniques). By analyzing and evaluating the physical trajectory, speed, and direction of subject motion in complex scenarios such as non-rigid motion, we aim to precisely predict and intervene at transition points where drastic motion changes occur, attempting to proactively address these complex edge cases.

---

> ### Author Response · Authors · 2025-11-24
> **Response to Reviewer 36Wi**
>
> > **Response to Weakness 3 (Missing References and Comparisons):**
>
> We thank the reviewer for pointing out these highly relevant works. First, we respectfully note that **Zhu et al. \[1] (Generative Inbetweening) was indeed cited and discussed in our original submission** (see Sec. 2 "Video Frame Interpolation" and Ref. \[1390] in the bibliography). Regarding **Zhang et al. \[2] (MoG)**, we acknowledge it as important concurrent work (arXiv, Jan 2025) and have included it in the revised discussion.
>
> While all three methods address consistency, AFB employs a fundamentally different paradigm:
>
> 1. vs **FCVG \[1]:** FCVG relies on extracting explicit conditions (matched lines/pose) and performing linear interpolation (or user-defined paths) to guide generation, which requires **fine-tuning extra modules**. Its geometric matching can fail when boundary frames differ significantly, and its linear assumption often leads to rigid motion in complex scenarios (e.g., dancing). In contrast, **AFB is training-free**. Instead of enforcing a pre-defined linear path, we utilize reverse generation to adaptively detect "semantic breakpoints" for anchor insertion. This enhances semantic guidance while preserving the base model's generative freedom for natural, non-linear motion.
>
> 2. vs **MoG \[2]:** MoG relies on optical flow warping and requires **fine-tuning spatial layers** to correct warping errors. Its performance is inherently bounded by the accuracy of optical flow estimation, which often fails in scenarios with drastic changes or severe occlusion. Furthermore, MoG is tailored for U-Net architectures (DynamiCrafter) and, as explicitly noted by its authors, currently lacks adaptation to modern DiT-based models. In contrast, **AFB avoids pixel-level optical flow dependency;** instead, it effectively mitigates information attenuation by bridging semantic gaps with high-quality anchor frames. Crucially, as a plug-and-play solution, AFB easily adapts to sota models (e.g., Wan2.1, Hunyuan).
>
> 3. We conduct quantitative and qualitative experiments on both methods using the dataset. Quantitative results are presented in the table below, and qualitative results are displayed in Appendix I Figure 20. Experiments demonstrate that our method outperforms both approaches in both quantitative and qualitative evaluations. Furthermore, our approach can generate 5-second (81-frame) videos, while FCVG and MoG are limited to generating only 1-second and 2-second short videos respectively.
>
> **Table: Quantitative comparison with related works.** AFB significantly outperforms both FCVG and MoG across all metrics.
>
> | Baseline Models | LPIPS $\downarrow$ | FVD $\downarrow$ | SSIM $\uparrow$ | PSNR $\uparrow$ |
> | :--- | :---: | :---: | :---: | :---: |
> | FCVG | 0.19 | 438.83 | 0.86 | 33.62 |
> | MoG  | 0.26 | 421.53 | 0.69 | 18.41 |
> | **Wan2.1 + AFB (Ours)** | **0.16** | **375.12** | **0.97** | **35.41** |
>
>
> * \[1] Zhu T, Ren D, Wang Q, et al. Generative inbetweening through frame-wise conditions-driven video generation\[C]//Proceedings of the Computer Vision and Pattern Recognition Conference. 2025: 27968-27978.
>
> * \[2] Zhang G, Zhu Y, Cui Y, et al. Motion-aware generative frame interpolation\[J]. arXiv preprint arXiv:2501.03699, 2025.
>
>
>
> > **Response to Weakness 4 (Visual Quality in Supplementary Materials):**
>
> We appreciate the feedback regarding visual evidence. We have updated the supplementary material with added visual comparisons and quantitative benchmarks against FCVG\[1] and MoG\[2].
>
> 1. **Quantitative Comparison:** As shown in the table, **AFB significantly outperforms both FCVG and MoG.** Specifically, AFB achieves the lowest FVD (375.12) and the  highest PSNR (35.41), demonstrating superior temporal coherence and visual fidelity.
>
> 2. **Visual Quality & Video Duration:** Firstly, we wish to point a key difference in task difficulty: **FCVG \[1] and MoG \[2] typically generate short clips (approx. 1-2 seconds).** In contrast, **AFB generates longer, complex videos (approx. 5 seconds)**. Despite the increased difficulty of generating longer sequences with complex dynamics, AFB still maintains better semantic consistency than the baselines. As detailed in the supplementary comparisons (Appendix I Figure 20), **Both FCVG and MoG exhibit noticeable ghosting artifacts on static objects, such as the cabinet handles (highlighted in red boxes).** In contrast, AFB leverages semantic anchors to generate clean, artifact-free frames with better logical coherence.
>
> * \[1] Zhu T, Ren D, Wang Q, et al. Generative inbetweening through frame-wise conditions-driven video generation\[C]//Proceedings of the Computer Vision and Pattern Recognition Conference. 2025: 27968-27978.
>
> * \[2] Zhang G, Zhu Y, Cui Y, et al. Motion-aware generative frame interpolation\[J]. arXiv preprint arXiv:2501.03699, 2025.

---

> ### Author Response · Authors · 2025-11-24
> **Response to Reviewer 36Wi**
>
> > **Response to Questioin1 (Optimization Plan for Reverse Pass):**
>
> Yes,  we have implemented an optimization plan based on "reducing denoising steps," as suggested.
>
> As detailed in our response to Weakness1 (Significant computational overhead) above, we develop the Fast-AFB strategy, which employs truncated denoising (terminating the reverse pass at step 15) and utilizes the predicted clean frame $\hat{x}_0$ for robust anchor selection. **This optimization successfully reduces the inference time overhead from 105% to 35%, while maintaining high semantic consistency and performance comparable to the full-pass method.** Please refer to the "Weakness 1" section above for the detailed methodology and experimental results.

---

### Official Review · Reviewer_QCjc · 2025-11-01

**Soundness:** 3
**Presentation:** 3
**Contribution:** 2
**Rating:** 4
**Confidence:** 4

**Summary:**

The paper presents Anchor Frame Bridging (AFB), which is an innovative method to improve first-last frame video generation (FLF2V). The approach addresses a major issue: ensuring that all the frames between the first and last are consistent both in terms of their semantic content and the timing of events. The key idea is to insert a high-quality "anchor frame" that acts like a bridge in between the start and end frames. This helps reduce distortion and keeps the motion smooth. The authors show through their experiments that their method improves the quality and consistency of videos better than existing advanced models on their built datasets.

**Strengths:**

+ The reverse generation process that selects the anchor frame is a smart way to tackle the problem of losing information as frames are generated.
+ This method significantly boosts video quality and consistency compared to advanced baseline models, backed by various metrics and thorough evaluations, including user feedback.
+ Since it's a plug-and-play module, it can be added to existing I2V diffusion models without needing any retraining, making it a very practical and broadly applicable tool.

**Weaknesses:**

+ Physical Causality Issues: The method generates frames in reverse order (last to first), but the model was trained with forward-time rules. This may create inconsistencies in scenarios with strong physical dynamics. For instance, reversing the video of ripples created by a stone drop doesn’t correctly model the physical event of ripples contracting to eject a stone. Although the authors’ datasets might not show this issue, it limits the method's use in situations with clear, irreversible physical laws.
+ Assumption of Symmetrical Nature: The method relies on the idea that the weakest frame in reverse generation lines up symmetrically with the weakest frame in forward generation. This assumption is crucial for the anchor frame placement but is not clearly proven in the paper. It’s possible that complexities in the start and end frames could disrupt this symmetrical failure point.
+ Computational Overhead: The method involves generating frames in two passes, which doubles the time for inference. The authors mention this in Section D.2, but say that this extra time is "acceptable" is arguable. Especially for models like CausVid, which need to operate with low latency and start playback before the video is fully ready. The requirement for a complete reverse-pass generation could undermine these models' real-time capabilities.

**Questions:**

The same to weaknesses.
1. Physical Causality: Could the authors show examples of scenarios with definite, irreversible physical laws like shattering glass, a splash, or smoke dissipating to better explain where their method works best?
2. Symmetrical Nature: Could there be an analysis or study to back up the symmetrical failure point assumption? Specifically, does this idea stand in different types of motion or when the complexity of the first and last frames vary greatly?

---

> ### Author Response · Authors · 2025-11-24
> **Response to Reviewer QCjc**
>
> We sincerely thank Reviewer for the constructive feedback and the positive comments. We are greatly encouraged that the reviewer recognizes our Anchor Frame Bridging (AFB) as an **"innovative method"** and a **"smart way"** to tackle the information attenuation problem in FLF2V tasks. We also appreciate the recognition of our method as a **"practical and broadly applicable tool"** that **"significantly boosts video quality and consistency"**  and without the need for retraining.  In the following, we provide detailed responses to the weaknesses and questions raised. We hope our response fully resolves your concerns.
>
>
> > **Response to Weaknesses 1 (Physical Causality Issues):**
>
> We thank the reviewer for this insightful observation. We agree that generating complete, time-reversed sequences of irreversible physical events (e.g., broken glass restoration) is not the primary objective of base I2V models, nor are such counter-intuitive examples prevalent in their training datasets.
>
> However, we wish to clarify that the effectiveness of AFB does not hinge on the perfection of the full-length reverse video. As long as the **initial segment** of the reverse sequence (i.e., frames close to the input frame $I_{N-1}$) maintains high image quality and semantic consistency, we can extract a valid anchor frame, which can effectively carry  accurate semantic guidance into the forward generation process, ensuring the final forward video remains both high-quality and physically coherent.
>
> To validate this, we conduct supplementary experiments on the suggested scenarios. **The qualitative experimental results are presented in Appendix F Figure 17 .** The results confirm that while full-length reverse videos indeed exhibit physical artifacts in later stages, **the initial segments maintain good fidelity**. Specific observations include:
>
> 1. **Orange dropping into water:** In the later stage of reverse generation, an unnatural water column accompanies the orange emerging from the water; however, the initial phase (orange rising and splash converging) is visually natural.
>
> 2. **Glass shattering:** In the later stage, residual shards unreasonably remain in the air as the cup lifts; yet, the initial reconstruction phase (shards gathering from the ground) preserves good quality.
>
> 3. **Water droplet ripples:** In the later stage, the droplet falls abruptly (violating reverse physics); but the initial process (ripples smoothing out) is highly realistic.
>
> 4. **Smoke dissipating:** In the later stage, a dense plume suddenly rushes in; conversely, the visual quality of the initial smoke state is well-maintained.
>
> In conclusion, experiments demonstrate that even in irreversible physical events，the initial segment of the reverse generation remains a highly credible semantic source, allowing AFB to extract a valid anchor frame from this interval and thereby guide coherent forward video generation.

---

> ### Author Response · Authors · 2025-11-24
> **Response to Reviewer QCjc**
>
> > **Response to Weakness 2（Assumption of Symmetrical Nature）:**
>
>
> We thank the reviewer for raising this critical question. First, we wish to clarify a conceptual distinction: the weakest frame in reverse generation and the weakest frame in forward generation are not "symmetrically mirrored,"  but rather exhibit a high degree of "positional consistency". Our work leverages this characteristic to find continuity breakpoints in reverse generation (i.e., the positions where breaks occur in forward generation), then utilizes mirror symmetry to identify corresponding frames in the reverse process as anchor frames for insertion into the forward generation.
>
> To verify the robustness of this assumption, we conduct a quantitative experiment across our dataset, encompassing diverse motion types such as asymmetric motion and drastic deformation. We calculate the frame-wise average LPIPS scores for both forward generation ($I_0 \to I_{N-1}$) and reverse generation ($I_{N-1} \to I_0$) across all videos. The statistical results (summarized in the table below and **detailed in Appendix G Figure 18** ) reveal that the quality breakdown point (peak LPIPS) for forward generation occurs at **Frame 56**, while for reverse generation, it occurs at **Frame 55**. The absolute deviation is merely **1 frame**. This provides strong statistical evidence that the "positional consistency" between forward and reverse generation is robust. Regardless of the video content and motion complexity, the point of quality collapse in the reverse generation consistently aligns with that of the forward generation.
>
> **Table: Comparison of average LPIPS scores (lower is better) between Forward ($I_0 \to I_{N-1}$) and Reverse ($I_{N-1} \to I_0$) generation across 436 videos.**
>
>
> | Frame ID | 0 | 5 | 10 | 15 | 20 | 25 | 30 | 35 | 40 | 45 | 50 | **55** | 60 | 65 | 70 | 75 | 80 |
> | :--- | :---: | :---: | :---: | :---: | :---: | :---: | :---: | :---: | :---: | :---: | :---: | :---: | :---: | :---: | :---: | :---: | :---: |
> | Forward | 0.03 | 0.055 | 0.06 | 0.07 | 0.07 | 0.076 | 0.08 | 0.085 | 0.09 | 0.103 | 0.11 | **0.163** | 0.14 | 0.126 | 0.088 | 0.079 | 0.067 |
> | Reverse | 0.04 | 0.06 | 0.059 | 0.063 | 0.069 | 0.074 | 0.079 | 0.084 | 0.087 | 0.127 | 0.179 | **0.22** | 0.152 | 0.152 | 0.135 | 0.114 | 0.086 |
>
>
> This alignment is not coincidental but is fundamentally driven by "information attenuation": **a systematic issue governed by model architecture and error accumulation, independent of generation direction.** Specifically, most FLF2V models are adapted from I2V architectures. Their inherent characteristic determines that the semantic constraints from boundary frames gradually decay over time (as analyzed in  Fig.1(a) of the main text).
>
> During forward generation, as the distance from the first frame increases, the semantic constraint weakens, causing minor visual flaws in each frame to go uncorrected. These tiny deviations **propagate, accumulate, and amplify (Error Accumulation)** frame-by-frame, eventually causing the generated content to deviate from a plausible trajectory and triggering a continuity break at a specific "semantic vacuum"( as shown in Fig.7).  Although reverse generation changes direction, the mechanisms of "constraint decay" and "error accumulation" persist, inevitably leading to a continuity break at a **similar position**. AFB precisely leverages reverse generation to localize this worst-quality breakdown point and inserts a high-quality anchor frame at the corresponding position in the forward process, thereby effectively mitigating information attenuation at that critical location.

---

> ### Author Response · Authors · 2025-11-24
> **Response to Reviewer QCjc**
>
> > **Response to Weakness 3（Computational Overhead）：**
>
> We thank the reviewer for pointing out the computational efficiency and the comparison with CausVid. We fully agree that AFB’s generation mechanism is not designed for low-latency streaming applications like CausVid.
>
> However, we wish to clarify that this distinction stems from the fundamental difference in task nature. CausVid utilizes block-wise causal attention, designed to generate video streams rapidly based on historical context. In contrast, FLF2V is a **dual-end constraint problem**: the generated video must not only continue the first frame but also precisely converge to a fixed last frame. Forcibly applying a CausVid-like streaming architecture to FLF2V would fail because the model generates early chunks without perceiving the future tail constraint, making Global Motion Trajectory Planning impossible. As generation proceeds, the established trajectory would inevitably face irreconcilable semantic or spatial conflicts with the fixed last frame.
>
> Nevertheless, we agree that improving computational efficiency is a crucial consideration for deployment. Therefore, we conduct new experiments and proposed an optimized **Fast-AFB** strategy.
>
> 1. **Methodology Improvement:** During the reverse generation process, we employ a **truncated denoising** approach to accelerate inference. In the early stages of reverse generation, the latent variable $x_t$ contains high noise, making direct LPIPS evaluation unreliable. To address this, **Fast-AFB** evaluates the **predicted clean frame** $\hat{x}_0$ derived from the current step $x_t$, rather than $x_t$ itself. The prediction formula is:
>                     $$\hat{x} _ 0 = \frac{x _ t - \sqrt{1-\bar{\alpha} _ t}\epsilon _ \theta(x _ t, t)}{\sqrt{\bar{\alpha} _ t}}$$
> 2. **Experimental Results:** As shown in the table below, reducing the stop step significantly lowers time overhead. **Notably, terminating at timestep 15 ($t=15$) achieves an optimal balance between performance and cost:** the FVD is 388.45, which is very close to the full-pass AFB (375.12) and still significantly outperforms both the original Wan2.1-I2V (449.68) and Wan2.1-FLF2V (413.68), while the inference time increases by only **35%** compared to the baseline (27 min vs. 20 min).
>
> **The Fast-AFB strategy offers users a flexible "Efficiency-Quality Trade-off":** users can prioritize maximum fidelity with the full AFB, or opt for Fast-AFB to achieve substantial performance gains over baselines with only a marginal increase in time cost.
>
> **Table1: Comparison of average LPIPS scores (lower is better) between Forward ($I_0 \to I_{N-1}$) and Reverse ($I_{N-1} \to I_0$) generation across 436 videos.**
>
> | Strategy | Metric | Stop Step ($K$) | Time (min) | Overhead | FVD $\downarrow$ | LPIPS $\downarrow$ | SSIM $\uparrow$ | PSNR $\uparrow$ | GPT-4o $\uparrow$ | Gemini $\uparrow$ |
> | :--- | :--- | :---: | :---: | :---: | :---: | :---: | :---: | :---: | :---: | :---: |
> | AFB | Q function(LPIPS) | 10 | 25 | +25% | 476.52 | 0.24 | 0.79 | 28.03 | 65.74 | 67.23 |
> | AFB | Q function(LPIPS) | 25 | 31 | +55% | 428.96 | 0.20 | 0.88 | 31.74 | 81.53 | 82.64 |
> | AFB | Q function(LPIPS) | 40 | 37 | +85% | 392.53 | 0.18 | 0.92 | 33.67 | 84.62 | 84.93 |
> | AFB | Q function(LPIPS) | 45 | 41 | +105% | 375.12 | 0.16 | 0.97 | 35.41 | 88.64 | 89.35 |
>
> **Table2: Performance of Fast-AFB strategy. At step 15, Fast-AFB achieves an optimal balance between performance and efficiency.**
>
> | Strategy | Metric | Stop Step ($K$) | Time (min) | Overhead | FVD $\downarrow$ | LPIPS $\downarrow$ | SSIM $\uparrow$ | PSNR $\uparrow$ | GPT-4o $\uparrow$ | Gemini $\uparrow$ |
> | :--- | :--- | :---: | :---: | :---: | :---: | :---: | :---: | :---: | :---: | :---: |
> | Fast-AFB | Q function(LPIPS) | 5 | 23 | +15% | 412.73 | 0.19 | 0.90 | 32.74 | 82.53 | 81.64 |
> | Fast-AFB | Q function(LPIPS) | **15** | **27** | **+35%** | **388.45** | **0.18** | **0.93** | **33.68** | **85.32** | **86.13** |
> | Fast-AFB | Q function(LPIPS) | 40 | 37 | +85% | 379.35 | 0.16 | 0.96 | 34.81 | 87.59 | 87.30 |

---

> ### Author Response · Authors · 2025-11-24
> **Response to Reviewer QCjc**
>
> > **Response to Question 1 ( Physical Causality):**
>
>  We conduct **supplementary experiments on the specific scenarios mentioned.** As detailed in our response to **"Physical Causality Issues"** in the Weaknesses section, we analyze cases including **orange dropping (splash), glass shattering, water ripples, and smoke dissipating**.
>
> Our observations confirm that while the full reverse videos in these irreversible scenarios exhibit logical artifacts in later stages, the **initial segments (close to the conditioning frame) remain highly realistic**. This demonstrates that our method **remains effective in these complex scenarios** by leveraging an anchor frame derived from this reliable initial interval to guide the generation.
>
> Please refer to the "Response to Weaknesses 1" section above for the detailed breakdown of each example.
>
> > **Response to Question 2 (Symmetrical Nature):**
>
> **1. Question 2-1: Validation of Symmetrical Assumption**
>
> Response: **Yes, we have validated this assumption through a large-scale quantitative experiment.** By calculating the **average** LPIPS curves across 436 videos, we found that the statistical breakdown points for forward and reverse generation **align closely** (Frame 56 vs. Frame 55 on average).This confirms that the "positional consistency" between forward and reverse generation is a robust statistical characteristic driven by systematic information attenuation. **For the detailed statistical table and theoretical explanation, please refer to Response to Weakness2 above.**
>
> **2. Question 2-2: Robustness to Motion and Complexity Variance**
>
> Response: to address this, we conduct supplementary evaluations **on three specific scenarios representing different challenges (visualizations provided in Appendix H Figure 19).** Our findings are as follows:
>
> 1.) **Complex Motion (e.g., Parkour):** Despite the highly dynamic nature of the motion, our method successfully identifies a valid anchor frame, generating coherent high-speed actions.
>
> 2.) **High Complexity Variance with Continuity (e.g., Character Scene Transition):** In cases where the starting and ending backgrounds differ significantly but the subject remains consistent (e.g., the same character walking from a **corridor to the outdoors**), AFB still functions effectively. The semantic continuity of the subject allows the anchor frame to bridge the complexity gap.
>
> 3.) **High Complexity Variance without Continuity (e.g., Flower $\rightarrow$ Girl):** In extreme cases where the first and last frames are semantically unrelated, the generated result resembles a **disjointed, slideshow-like transition** rather than a coherent video. However, we wish to clarify that the primary objective of AFB is to preserve physical and semantic continuity; therefore, **semantically unrelated boundary frames fall outside the scope of our current method.**  While AFB is not designed to resolve this semantic disjointedness, addressing such non-continuous constraints remains an interesting direction for future work.

---

### Author Response · Authors · 2025-12-02
**Response to ACs**

To the ACs:

We sincerely thank ACs for their outstanding effort when dealing with our paper.

In this work, we reveal the **"information attenuation"** existing in the **First-Last Frame Video Generation (FLF2V)** scenario, where guidance from boundary frames weakens towards the middle of the sequence. Furthermore, we propose **Anchor Frame Bridging (AFB)**, a training-free and plug-and-play framework that explicitly bridges semantic gaps, thereby addressing the critical challenges of semantic degradation and temporal inconsistency in FLF2V.
***
According to the following review comments, our proposed **Anchor Frame Bridging (AFB)** holds significant potential implications for **First-Last Frame Video Generation (FLF2V)** in the context of controllable video generation:

+ **The work demonstrates significant novelty and practicality.** (`All Reviewers`)
+ **The proposed methodology is praised as "novel and clever", an "elegant, non-trivial solution", an "innovative method", and a "methodological innovation"** that effectively addresses information attenuation. (`Reviewer zLy5, QCjc, hmuv`)
+ **The approach features a "practical, training-free design"** that is "critical for real-world FLF2V applications", with "significant practical advantages" and serves as a "broadly applicable tool".(`Reviewer 36Wi, zLy5, QCjc`)
+ **The study presents rigorous evaluation**, including "rigorous validation", "targeted problem diagnosis", and "empirical rigor", building a "robust case" for the method's effectiveness. (`Reviewer 36Wi, hmuv, zLy5`)
***
We have appropriately addressed all concerns of the reviewers, including:
*   **Computational efficiency** (`Reviewer QCjc`-W3; `Reviewer 36Wi`- W1&Q1; `Reviewer zLy5`- W1&Q1)
*   **Justification of the "Mirror Heuristic"** (`Reviewer QCjc`-W2 & Q2; `Reviewer zLy5`-W3 & Q3; `Reviewer hmuv`-W1)
*   **Scalability of our method** (**multi-anchor performance**, `Reviewer zLy5`-W2 & Q2; `Reviewer 36Wi`-W2; **scalability and robustness in edge cases**, `Reviewer hmuv`-W3; `Reviewer QCjc`-Q2-2).
***
We expect ACs to fully consider the following factors when making the final decision:

1.  **A highly effective, training-free, and plug-and-play solution**, along with sound justifications, remarkable robustness and scalability confirmed by all reviewers.
2.  **Comprehensive responses** to all reviewer comments.
3.  **The demo and implementation code** opened to the reviewers (in the supplementary material).

Best regards,

The Authors

---

### Meta-Review · Area_Chair_NdYH · 2026-01-11

**Summary:**

Reviews were initially borderline (4,4,6,6), with reviewers complaining about computational overhead, missing baselines, heuristics, scalability, and limitations. The authors provided a very comprehensive rebuttal that seems to have addressed all the concerns.

**Reviewer Concerns:**

Reviewers raised many concerns, but it seems all of them were addressed (or limitations were conceded) in the author rebuttal. A full list below:

- Concerns that generating frames in reverse (last-to-first) violates physical laws in irreversible events
- Doubts about the mirroring heuristic
- Concerns about hyperparameter sensitivity
- Concerns about computational overhead
- Edge cases like non-rigid motion or occlusions
- Missing baselines
- Dataset size and diversity
- Visual quality

**Reviewer Scores:**

Scores started as (4,4,6,6), but the authors have spent a considerable amount of time rebutting the reviewer concerns, and I would expect one or both of the borderline-negative reviews to have increased in score, resulting in a (6,6,6,6) or (4,6,6,6). As such, I would expect this submission to end with a positive score.

---

### Decision · Program_Chairs · 2026-01-26

Accept (Poster)